# UXT chaperone prevents proteotoxicity by acting as an autophagy adaptor for p62-dependent aggrephagy

Min Ji Yoon [1,4], Boyoon Choi[2,4], Eun Jin Kim[1], Jiyeon Ohk[2], Chansik Yang[1], Yeon-Gil Choi[1], Jinyoung Lee[1], Chanhee Kang[3], Hyun Kyu Song [1], Yoon Ki Kim [1], Jae-Sung Woo[1], Yongcheol Cho [1], Eui-Ju Choi[1], Hosung Jung [2✉] & Chungho Kim [1✉]

p62/SQSTM1 is known to act as a key mediator in the selective autophagy of protein aggregates, or aggrephagy, by steering ubiquitinated protein aggregates towards the autophagy pathway. Here, we use a yeast two-hybrid screen to identify the prefoldin-like chaperone UXT as an interacting protein of p62. We show that UXT can bind to protein aggregates as well as the LB domain of p62, and, possibly by forming an oligomer, increase p62 clustering for its efficient targeting to protein aggregates, thereby promoting the formation of the p62 body and clearance of its cargo via autophagy. We also find that ectopic expression of human UXT delays SOD1(A4V)-induced degeneration of motor neurons in a *Xenopus* model system, and that specific disruption of the interaction between UXT and p62 suppresses UXT-mediated protection. Together, these results indicate that UXT functions as an autophagy adaptor of p62-dependent aggrephagy. Furthermore, our study illustrates a cooperative relationship between molecular chaperones and the aggrephagy machinery that efficiently removes misfolded protein aggregates.

[1] Department of Life Sciences, Korea University, Seoul, Republic of Korea. [2] Department of Anatomy, Graduate School of Medical Science, Brain Korea 21 Project, Yonsei University College of Medicine, Seoul, Republic of Korea. [3] School of Biological Sciences, Seoul National University, Seoul, Republic of Korea. [4] These authors contributed equally: Min Ji Yoon, Boyoon Choi. ✉email: hosungjung@yonsei.ac.kr; chungho@korea.ac.kr

Newly synthesized proteins must go through a process that ensures proper folding, and this phenomenon is hampered by mutations, oxidative stress, or transcriptional/translational errors[1]. Misfolding of a protein often leads to the exposure of the hydrophobic regions that would otherwise be internally buried, and consequently, protein aggregation. Accumulation of protein aggregates ultimately results in cellular toxicity that postmitotic cells such as neurons are more vulnerable to. For example, protein aggregates containing amyloid-beta, huntingtin, and mutant form of superoxide dismutase 1 (SOD1) are respectively associated with neurodegenerative diseases, such as Alzheimer's disease[2,3], Huntington's disease[4–6], and a form of familial amyotrophic lateral sclerosis (ALS)[3,7]. The ubiquitin–proteasome system (UPS) and autophagy–lysosomal system are the two major surveillance systems that cooperatively detect and remove misfolded proteins[8]. The UPS recognizes and ubiquitinates the hydrophobic residues on the surface of misfolded proteins by ubiquitin ligases with the aid of molecular chaperones[9–11]. When the UPS is impaired or overloaded, misfolded proteins accumulate and often form intracellular aggregates that are then directed toward the alternative autophagy pathway[12,13].

During aggrephagy, or the selective degradation of protein aggregates by autophagy, protein aggregates are recognized by various autophagy receptors, which deliver them to the phagophore (the forming autophagosome)[14–16]. A well-known receptor for autophagy, p62 (also known as sequestosome-1), sequesters protein aggregates in condensates called the p62 body before eventually targeting them to the phagophore for proteolytic degradation[15,17,18]. As its role suggests, p62 contains multiple domains that are required for its interaction with proteins in diverse pathways (Fig. 1a). The N-terminal PB1 domain mediates homo-oligomerization that facilitates p62 body formation, and may increase its affinity for protein aggregates and/or the phagophore[18,19]. The C-terminal ubiquitin-associated (UBA) domain interacts with ubiquitinated protein aggregates[15,20] and the LC3-interacting region motif directly binds to Atg8/LC3 on the phagophore surface[17], thereby linking the protein aggregates to the phagophore membrane to form the autophagosome[17,21]. A pioneering study established the importance of PB1 and UBA domains in autophagy[15] by showing that a p62 deletion mutant that only contained PB1 and UBA domains can induce p62 body formation. The Keap interaction region domain mediates Keap1 binding when phosphorylated at Ser349 residue, releasing a transcription factor Nrf2 from Keap1 to control the gene expression[22] and/or modulating p62 activity by ubiquitination[23]. In contrast, although new proteins that regulate p62 function and interact with the zinc finger ZZ domain, LIM protein-binding domain (LB), and TRAF6-binding domain have recently been identified, the roles of these domains in p62-mediated autophagy remain relatively unclear[24–26].

In this study, we used an unbiased yeast two-hybrid approach to identify a possible regulator of p62 that binds to those not-well-characterized regions and identified UXT, a 157 amino acid long protein known to be ubiquitously expressed in various tissues[27], as a protein binding to the LB domain of p62. Among several functions of UXT suggested so far, including a cofactor of nuclear receptors[28,29], a regulator of NF-κB signaling[30], and a prefoldin-like molecular chaperone[30], we focused on its possible function as the chaperone. Prefoldin is a jellyfish-like hexameric chaperon[31] preventing protein aggregates found in various neurological diseases[32–34], presumably by delivering the aggregate-prone proteins to chaperonins, such as TRiC[35]. In an analogy to the function of prefoldin, we hypothesized that UXT may have a similar but distinct role, delivering protein aggregates to autophagy machinery by interacting with p62. Here, we show that UXT, presumably by forming an oligomeric form, can associate with protein aggregates to enhance p62 body formation. This results in efficient clearance of the p62 cargo by autophagy and amelioration of loss of motor function associated with an animal model of ALS in a p62-dependent manner. Based on these results, we propose that UXT may serve as a potential therapeutic target for drugs aimed at treating neurodegenerative diseases associated with protein aggregation, including ALS.

## Results

**A p62 interacting protein UXT colocalizes with protein aggregates.** The formation of p62 bodies, characterized by clear punctate structures that colocalize with aggregates, is thought to be dependent on the interaction between ubiquitinated protein aggregates and the UBA domain of p62, as well as PB1 domain-mediated oligomerization of p62 (ref. [15]). However, when a p62 mutant p62(ΔZZ–LB), lacking domains seemingly nonessential for p62 body formation, i.e., ZZ and LB domains (Fig. 1a), was transiently transfected for testing its ability to form the p62 body in p62 knockout HeLa cells (HeLa/p62KO; Supplementary Fig. 1a–e), we observed that p62 body formation by the mutant is not as efficient as that by wild type (WT): the p62 bodies formed by the mutant were smaller in size and number compared to those formed by the WT protein (Fig. 1b). For quantitative analysis of p62 body formation, we developed an in-house MATLAB code (Supplementary Code 1) that automatically selects pixels with fluorescence intensities exceeding three standard deviations of the mean of whole-cell intensity as regions of p62 bodies. It was also able to calculate a clustering index, defined as the sum of fluorescence intensity of p62 staining in the selected cell, divided by the mean of whole-cell fluorescence intensity (Supplementary Fig. 2a, b). In this unbiased analysis, the mutant p62 had a significantly lower clustering index than the WT (Fig. 1c). Consistently, the mutation also led to decrease in the number and size of p62 bodies (Supplementary Fig. 2c, d), suggesting that these domains may contribute to p62 body formation. HeLa/p62KO cells stably transfected with p62(ΔZZ–LB) also showed the similar defect (Supplementary Fig. 2e–h). Interestingly, a previous study had reported that p62 mutant lacking exons 4 and 5 do not bind to SOD1(A4V) associated with familiar ALS[36]. As the exons 3 through 5 encode the ZZ and LB domains, this is consistent with our findings and is also additional evidence that shows the importance of these domains in p62 body formation. To gain a mechanistic understanding of the function of these two domains, we performed yeast two-hybrid screens using the ZZ and LB domains as bait (Fig. 1a). In this screen, we found that 9 out of 27 URA3 and ADE2 reporter-positive clones were UXT (Supplementary Table 1). Notably, while most other URA3 and ADE2-positive clones were negative for the lacZ reporter assay, the UXT clones that were identified were positive (Fig. 1d and Supplementary Table 1), demonstrating their relatively strong affinity for p62. HA-tagged p62 and GFP-tagged UXT transiently expressed in HEK239T cells (Fig. 1e), as well as endogenous p62 and UXT in HeLa cells (Supplementary Fig. 2i) also co-immunoprecipitated, validating the interaction. A proximity ligation assay (PLA) also showed an interaction between endogenously expressed p62 and UXT in HeLa cells (Fig. 1f, g). We subsequently used p62 deletion mutants to map the region required for UXT binding to the LB domain (Fig. 1h and Supplementary Fig. 2j), especially from 184th to 204th residues (Fig. 1i, j and Supplementary Fig. 2k). The better binding of UXT to the ΔZZ mutant than to the WT (Fig. 1j) suggests that the neighboring ZZ domain may sterically hinder the interaction. To determine if UXT is involved in p62-mediated autophagy, we first utilized HeLa cells stably expressing the well-known aggregate-prone cystic fibrosis transmembrane conductance regulator

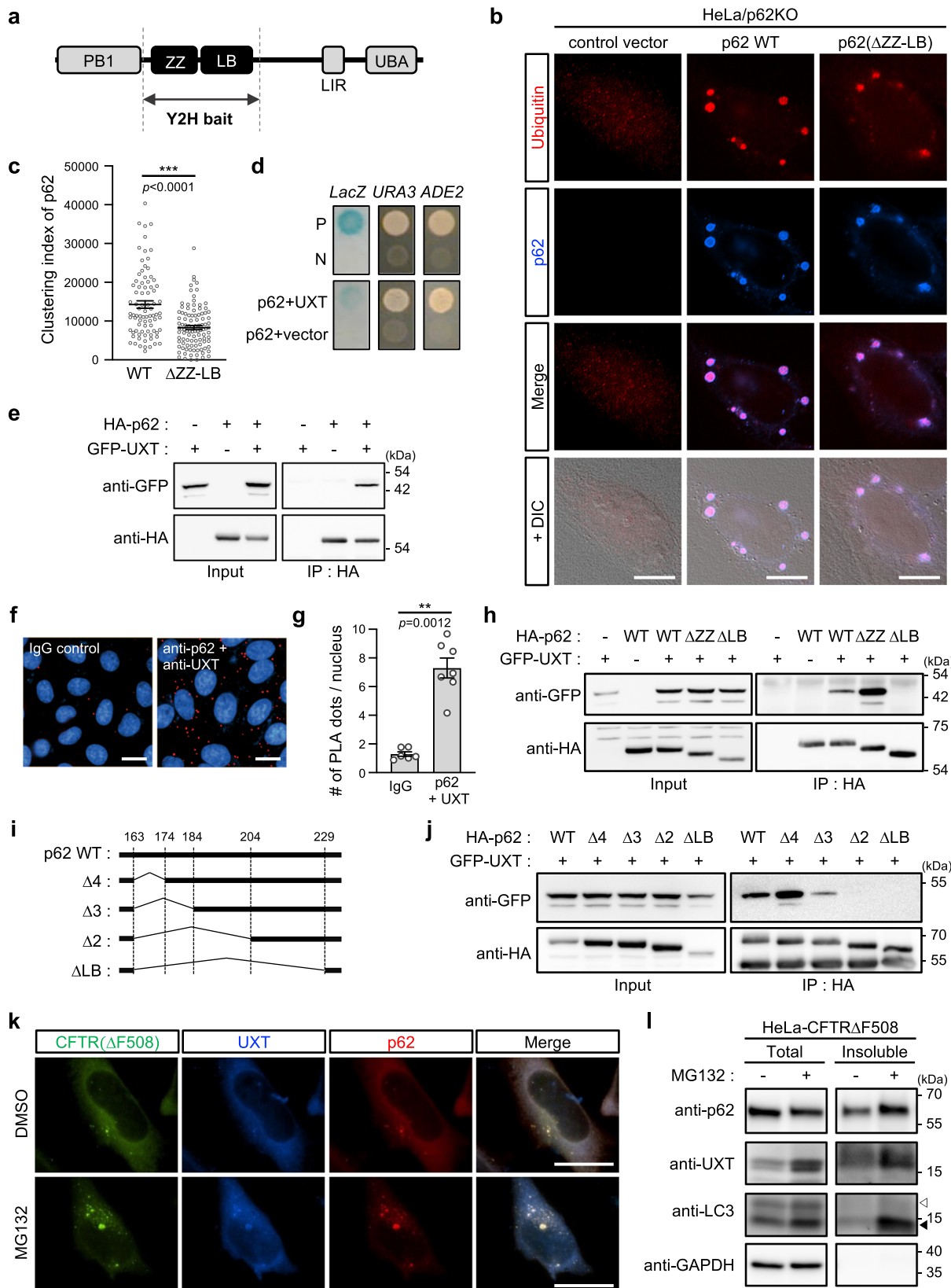

(CFTR) mutant CFTR(ΔF508)[37,38], and checked for colocalization of the CFTR aggregates with UXT and p62. As the UPS is responsible for degradation of the misfolded CFTR protein[38], inhibition of UPS by MG132 treatment increased accumulation of CFTR aggregates (Fig. 1k). In the aggregates, we could detect both p62 as well as UXT (Fig. 1k), suggesting that interaction between

p62 and UXT may play a role in protein aggregates. The targeting of UXT to protein aggregates upon MG132 treatment was also confirmed by increased UXT levels in the protein aggregates containing detergent-insoluble fraction, in which increase of autophagic machinery, p62 and phosphatidylethanolamine (PE)-conjugated LC3 form (LC3-II)[39], was observed (Fig. 1l, bottom).

**Fig. 1 A p62 binding protein, UXT colocalizes with protein aggregates. a** Domain structure of p62. **b** p62 knockout HeLa cells (HeLa/p62KO) were transiently transfected with FLAG-tagged p62 (wild type or ΔZZ-LB). Cells were treated with 5 μM MG132 for 6 h, and stained with anti-ubiquitin and anti-p62 antibodies followed by anti-rabbit IgG-TRITC and anti-mouse IgG-Alexa-350 antibodies. DIC differential interference contrast images. Scale bar, 10 μm. **c** Clustering indexes (Supplementary Fig. 2) of p62 wild type or ΔZZ-LB mutant-transfected cells are calculated and represented as scatter plots ($n = 82$ and 101 cells examined over three independent experiments, mean ± SEM). ***$p < 0.001$ (two-sided Mann–Whitney test). **d** Yeast two-hybrid assays showing the interaction between the ZZ-LB domain of p62 and UXT using LacZ, URA3, and ADE2 reporters. P and N, positive and negative control, respectively. **e** HEK293T cells were transfected with HA-p62 and GFP-UXT, and their interaction was demonstrated by detection of GFP-UXT in precipitates of HA-p62. **f** Proximity ligation assay (PLA) dots (red) representing close proximity of antibodies against p62 and UXT (right) were increased compared to isotype control mouse and rabbit IgG antibodies (left). Nuclei were stained with DAPI (blue). Scale bar, 20 μm. **g** Number of PLA dots per nucleus is shown as bar graphs ($n = 6$ and 7 random fields for each condition over three independent experiments, mean ± SEM). **$p < 0.01$ (two-sided Mann–Whitney test). **h** Interaction between UXT and p62 deletion mutants was analyzed as in **e**. **i** Diagram of p62 LB domain-deletion mutants. **j** Interactions between UXT and p62 mutants were analyzed as in **e**. **k** HeLa cells stably expressing GFP-CFTR(ΔF508) were treated with 2.5 μM MG132 for 12 h, and localization of the CFTR mutant and endogenous UXT (stained with anti-UXT antibody followed by anti-mouse IgG-Alexa-350) and p62 (stained with anti-p62 antibody followed by anti-rabbit IgG-TRITC) were analyzed and compared to the untreated control. Scale bar, 25 μm. **l** Under the same conditions used for **k**, the levels of UXT and p62 protein in the RIPA-insoluble fraction were analyzed. White and black arrowheads indicate LC3-I and -II, respectively.

Notably, the levels of total UXT protein were also increased in MG132-treated cells (Fig. 1l, top), suggesting that UXT may play a role in conditions with increased misfolded protein aggregates.

**UXT overexpression protects DRG neurons from proteasome inhibition-induced cell death.** Colocalization of UXT with protein aggregates led us to propose that UXT may have a role in the handling of protein aggregates. As neuronal cells are known to be more vulnerable than other types of cells to protein aggregates[40], we tested the effect of UXT on protein aggregation condition in primary dorsal root ganglion (DRG) neurons. We found that UXT colocalizes with p62 bodies in DRG neurons, and as expected, proteasome inhibition by MG132 treatment increased p62 body formation (Fig. 2a). This was similar to what we observed in HeLa cells, most p62 bodies colocalized with protein aggregates stained with anti-ubiquitin (Fig. 2a, upper panels) and with UXT (Fig. 2a, lower panels). We next investigated the consequences of proteasome inhibition in DRG neurons and the effect of increased expression of UXT in the process. To this end, we transduced DRG neurons with lentivirus bearing mCherry-fused UXT or mCherry control, treated them with MG132, and observed neurons over time under a fluorescence microscope equipped with temperature and humidity controllers (Fig. 2b). At ~8 h after 2.5 μM MG132 treatment, cell bodies of the DRG neurons started to shrink first and generates blebs that eventually burst within ~2.5 h after the first appearance of blebbing (Fig. 2c and Supplementary Movie 1a, b). The cells showed blebbing in the real-time observation was also specifically stained with anti-cleaved caspase-3 (Supplementary Movie 1c), an apoptosis marker[41], suggesting that the cell death may be mediated by apoptosis. As such, we used bursting as a marker of cell death to study survival. We found that DRG neurons transduced with UXT were more resistant to MG132-induced toxicity than control DRG neurons, with 50% cell death observed at 1600 and 900 min, respectively, after MG132 treatment (Fig. 2d). No cell death was observed within the same time window for the untreated control (Fig. 2d, blue line), indicating that cell death observed was due to MG132-induced toxicity. Together, these results suggest that UXT may exert protective functions against protein aggregate-induced neuronal cell death.

**UXT cooperates with p62 during autophagy.** As UXT directly interacts with the autophagy receptor p62, we hypothesized that the protective effect of UXT can be attributed at least in part to p62-dependent autophagy. Indeed, when autophagy was blocked using an autophagic flux inhibitor (Fig. 3a, b), bafilomycin A142,

or transient p62 knockdown (Fig. 3c), the accumulation of UXT protein was observed (Fig. 3a, b), suggesting its possible involvement in p62-dependent autophagy. Blocking UPS by MG132—which increased the UXT protein level (Figs. 1l and 3a)—in this background further enhanced UXT protein accumulation (Fig. 3a). As we did not detect significant differences in UXT mRNA levels under these treatment conditions by quantitative reverse transcription polymerase chain reaction (RT-PCR; Supplementary Fig. 3a), the increase in UXT protein levels does not appear to be due to increase in mRNA expression. Instead, the accumulation of UXT protein with MG132 treatment suggests that UXT is constitutively degraded by UPS, but stabilized under an aggregation-prone condition by an unidentified mechanism for its possible roles in autophagy. To test this hypothesis, we investigated if UXT is recruited to the autophagosome. When autophagy was induced using amino acid deprivation along with inhibition of autophagic flux in HeLa cells that stably expressed GFP-LC3, we observed an increase in the number of puncta positive for LC3 and p62 (Fig. 3d), which are the markers of autophagosomes[42]. The number of UXT puncta with higher fluorescence intensity was also increased in cells, in which autophagy was induced compared to that in untreated control cells (Fig. 3d). Under the autophagy-induced conditions, most of the UXT-positive puncta colocalized with p62, and some of the UXT- and p62-positive puncta also colocalized with LC3 (Fig. 3d), suggesting that UXT in combination with p62 can be targeted to the phagophore. Quantification by using Pearson's correlation coefficient showed a significant increase in the colocalization within UXT, p62, and LC3 (Fig. 3e and Supplementary Fig. 3b). Moreover, when we attempted to purify UXT (UXT-eYFP-10×His) expressed in HEK293E cells by sequential Ni-NTA and size-exclusion chromatography (SEC; Fig. 3f), we could see the UXT-containing fractions observed near the column void volume as a large complex contain various binders (Fig. 3g) of which we could detected p62 and, based on the molecular weight, LC3-II by western blots (Fig. 3h), suggesting UXT's ability to autophagosome targeting.

**UXT mediates the clearance of aggregates via autophagy.** As UXT appeared to be targeted to the LC3-positive autophagosomes, we investigated if UXT could induce the clearance of aggregates via autophagy. We first transfected HEK293T cells with GFP-tagged SOD1 or its aggregate-prone mutant, SOD1 (A4V), and also co-transfected the cells with mCherry-fused UXT to determine colocalization. When observed under a fluorescence microscope, WT SOD1 was diffused throughout the cytosol, while SOD1(A4V) was mostly condensed in the form of one or

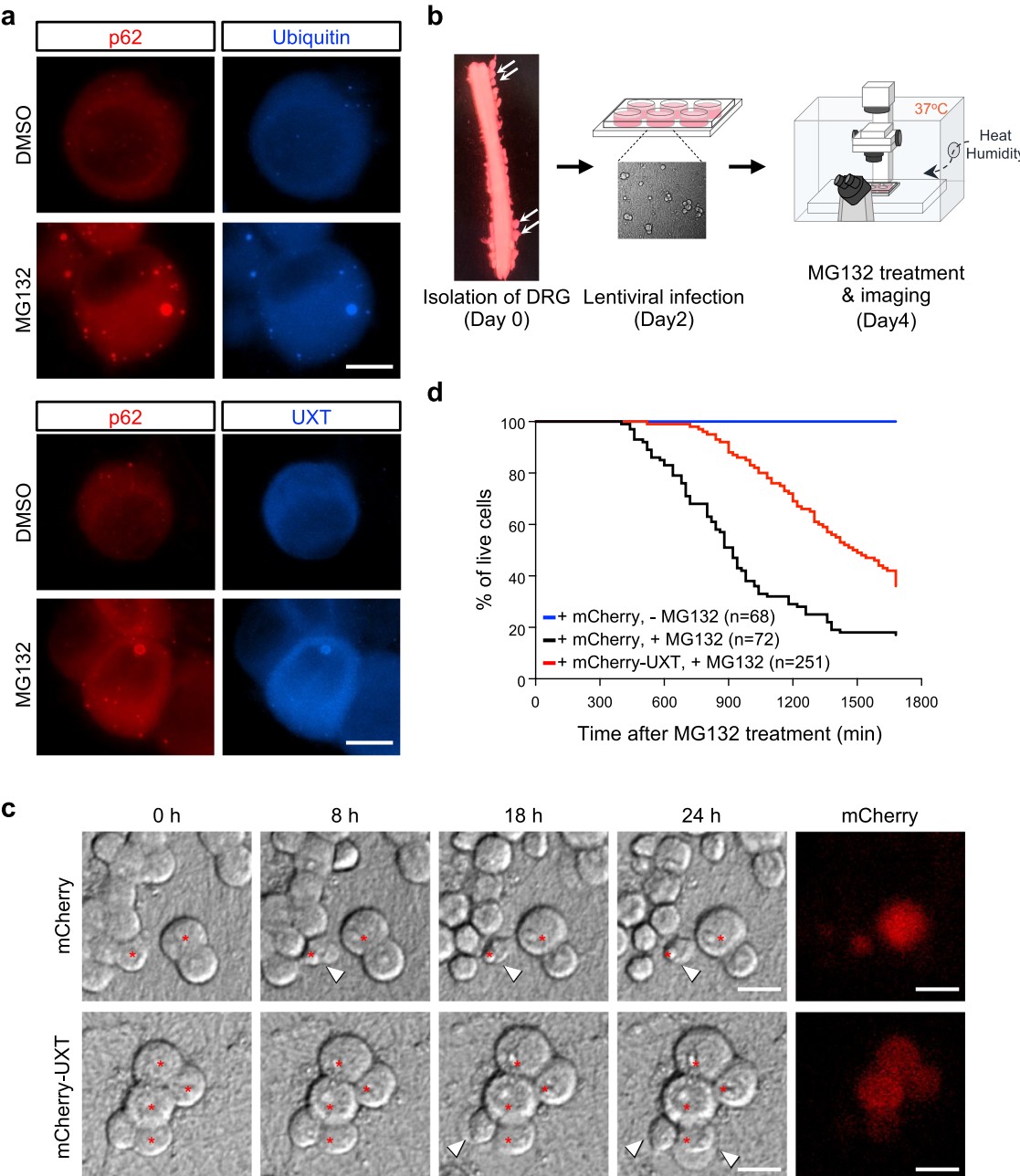

**Fig. 2 UXT overexpression protects neuronal cells from proteasome inhibition-induced cell death. a** Immunocytochemistry of dorsal root ganglion (DRG) neurons. DRG neurons were treated with 2.5 μM MG132 for 12 h and compared to untreated controls. Endogenous p62 and ubiquitin-positive protein aggregates (upper panels) or p62 and UXT (lower panels) were stained under both conditions with their specific primary antibodies, and subsequently with anti-rabbit IgG-TRITC and anti-mouse IgG-Alexa-350 as secondary antibodies. Scale bar, 10 μm. **b** On day 0, DRG neurons (white arrows) were isolated from day 13.5 mouse embryos and cultured for 48 h. On day 2, mCherry or mCherry-UXT was introduced into DRG neurons by lentiviral transduction and cells were cultured for 2 days. On day 4, the DRG neurons were treated with 2.5 μM MG132 and observed under a live imaging microscope with temperature and humidity control. DIC (differential interference contrast) images were collected for 28 h after MG132 treatment at 20 min intervals (Supplementary Movie 1). To identify the transduced cells, fluorescence images were also taken at the end. **c** DIC images collected at 0, 8, 18, and 24 h and mCherry fluorescence images of DRG neurons are shown. Transduced cells are indicated with red asterisks. Membrane blebs are indicated with arrowheads. Scale bar, 20 μm. **d** Survival of DRG neurons transduced with mCherry without MG132 treatment (blue), mCherry with MG132 treatment (black), and mCherry-UXT with MG132 treatment (red) was analyzed and is shown with the number of DRG neurons analyzed. This analysis was repeated twice with the essentially same results.

two aggregates (Fig. 4a and Supplementary Fig. 3c). UXT showed relatively dispersed punctate structures in control cells. However, in the SOD1(A4V)-transfected cells, UXT showed a dense pattern in close proximity to SOD1(A4V) aggregates (Fig. 4a and Supplementary Fig. 3d). Consistent with this and previous reports[43–45], SOD1(A4V) was found in the detergent-insoluble fraction of the cell lysate, and the amount of SOD1(A4V) protein aggregates was significantly reduced in UXT co-transfected cells (Fig. 4b, c). Inhibition of autophagy by bafilomycin A1 largely suppressed UXT-mediated reduction in detergent-insoluble SOD1(A4V) aggregates (Fig. 4, c), suggesting that UXT-dependent clearance of SOD1(A4V) aggregates occurred in

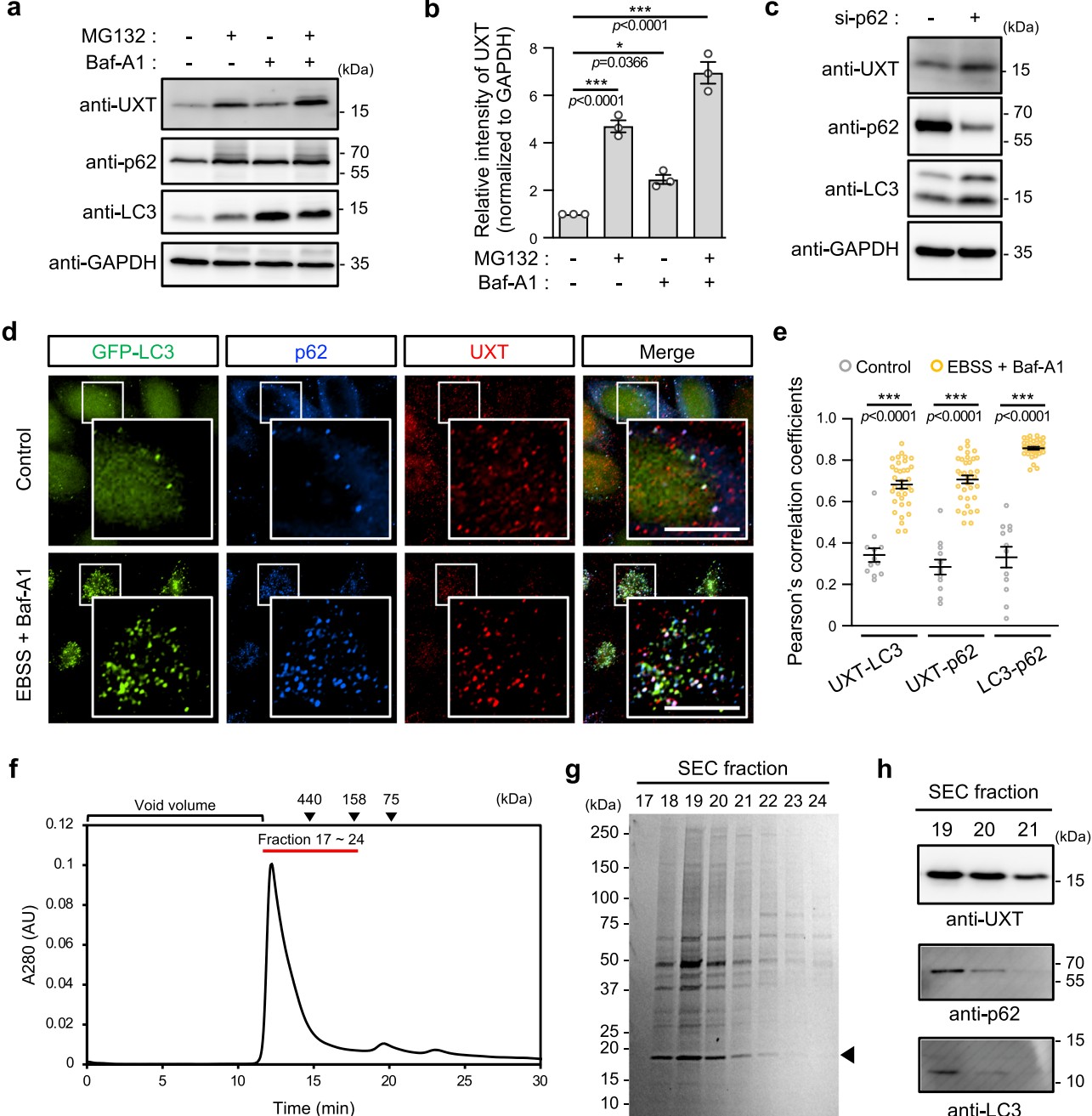

**Fig. 3 UXT and p62 cooperate and are involved in autophagy. a** HeLa cells were treated with 2.5 μM MG132 and/or 50 nM bafilomycin A1 (Baf-A1) for 12 h, and the whole-cell lysates were subjected to a western blot analysis. **b** Band intensities of UXT in each condition were normalized to that of GAPDH, and their relative intensities against non-treated condition are shown as bar graphs (*n* = 3, mean ± SEM). \**p* < 0.05; \*\**p* < 0.01; \*\*\**p* < 0.001 (one-way ANOVA using Bonferroni's multiple comparison test). **c** HeLa cells were transfected with p62-siRNA and then whole-cell lysates were subjected to a western blot analysis. **d** HeLa cells stably expressing GFP-LC3 were starved of amino acids using EBSS media, and this was followed by inhibition of the autophagic flux by treatment with 50 nM Baf-A1 for 4 h. The localization of GFP-LC3 (green), p62 (blue), and UXT (red) were analyzed by immunocytochemistry, using anti-p62 and anti-UXT as primary antibodies, and anti-rabbit IgG-Alexa-350 and anti-mouse IgG-TRITC as secondary antibodies. Scale bar, 10 μm. **e** Pearson's correlation coefficients between each protein were shown as scatter plots (*n* = 12 and 34 cells for control and EBSS + Baf-A1, respectively, mean ± SEM). \*\*\**p* < 0.001 (two-sided Mann–Whitney test). **f** UXT-eYFP-10×His construct was transfected into HEK293E cells, and the protein purified using a Ni-NTA column was treated with His-tagged HRV3C protease to cleave the region between UXT and eYFP. Resulting reaction was passed through the second Ni-NTA column and further separated by size-exclusion chromatography (SEC). **g** Coomassie blue-stained SDS–PAGE gel profile of UXT-containing SEC fractions in **f** is shown. Arrowhead indicates UXT. **h** The SEC fractions were further analyzed with western blot using anti-UXT, anti-p62, and anti-LC3 antibodies.

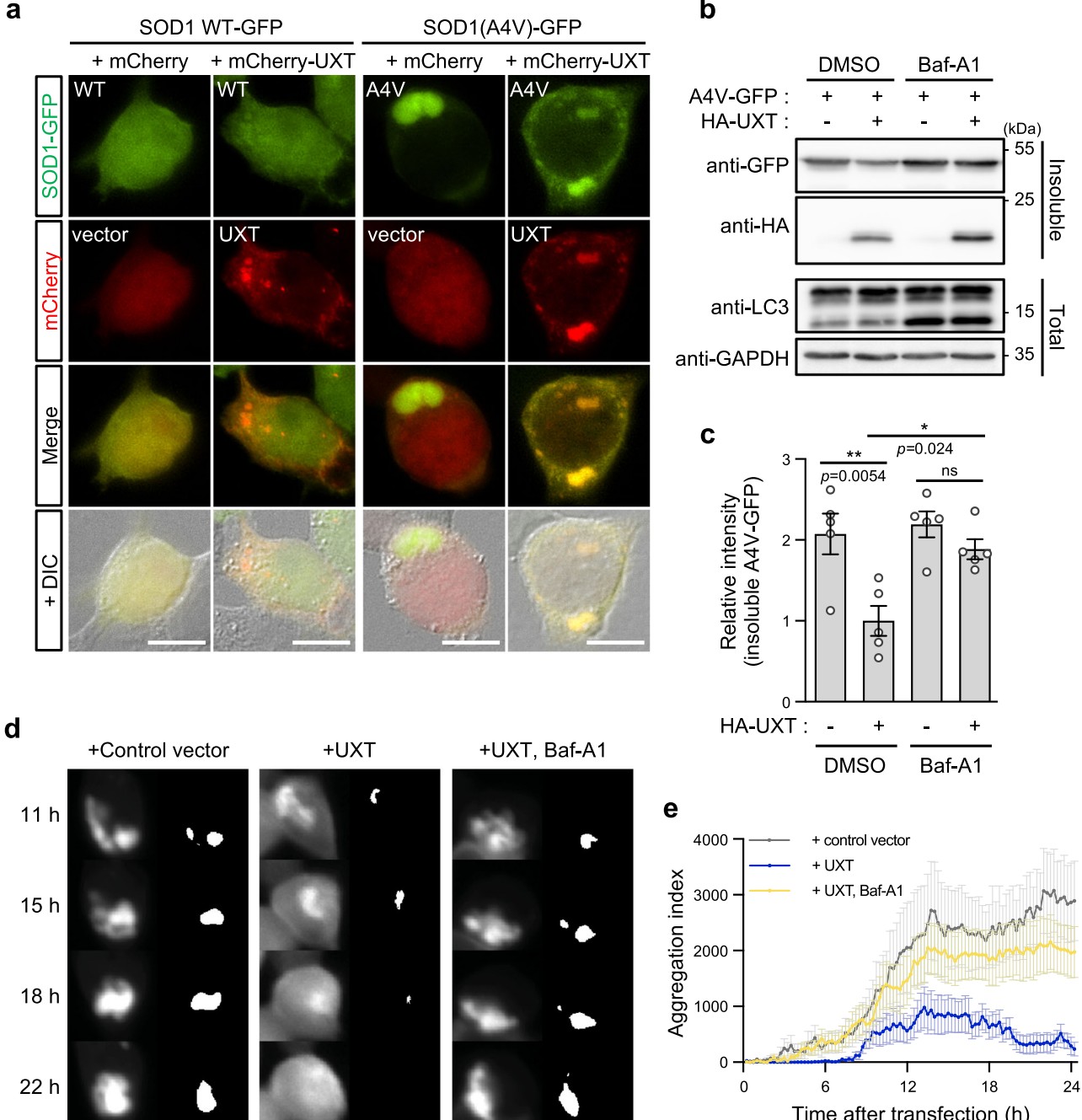

**Fig. 4 UXT mediates the clearance of SOD1(A4V) aggregates by autophagy. a** HEK293T cells were transiently transfected with SOD1 (wild type or A4V)-GFP and mCherry-UXT or mCherry-only control. Representative images of transfected cells, merged composites and differential interference contrast (DIC) images are shown. Scale bar, 10 μm. **b** HEK293T cells were transiently transfected with SOD1(A4V)-GFP and HA-UXT or vector, and were treated with 50 nM bafilomycin A1 (Baf-A1) for 12 h, as shown. SOD1 and UXT were detected by western blotting in the detergent-insoluble fraction. LC3 in whole-cell lysates were analyzed to confirm inhibition of autophagic flux by Baf-A1, and GAPDH was used as an internal loading control. **c** Band intensities of SOD1(A4V)-GFP in detergent-insoluble fractions were normalized against those of HA-UXT-transfected cells without Baf-A1 treatment, and are shown as bar graphs (*n* = 5, mean ± SEM). **p* < 0.05; ***p* < 0.01; ns not significant (one-way ANOVA using Bonferroni's multiple comparison test). **d** Immediately after HEK293T cells were transiently co-transfected with SOD1(A4V)-GFP and mCherry-UXT or mCherry-only control, the cells were either treated with 50 nM bafilomycin or DMSO as a control, and fluorescence images of SOD1(A4V)-GFP in mCherry-positive cells were captured at 15 min intervals for 24 h. Representative images at 11, 15, 18, and 22 h are shown. Black and white images indicating pixels automatically recognized as aggregate regions by an in-house MATLAB code (Supplementary Code 1) are also shown. Scale bar, 10 μm. **e** Averages of the aggregation indexes of cells in **d** that were vector control transfected (gray, *n* = 10), mCherry-UXT transfected (blue, *n* = 11), and mCherry-UXT transfected along with bafilomycin A1 (Baf-A1) treatment (yellow, *n* = 17) are shown. Mean ± SEM is indicated.

an autophagy-dependent manner. To better understand the role of UXT in protein degradation, we monitored expression and aggregates formation in HEK293T cells transfected with GFP-tagged SOD1(A4V), using a live cell imaging system (illustrated in Fig. 2b). SOD1(A4V) aggregated initially in the form of several small clusters that eventually coalesced to form dense aggregates (Supplementary Movie 2). However, in UXT-transfected cells, SOD1(A4V) did not form such dense aggregates, and growing aggregates quickly disappeared. Notably, the inhibition of autophagy by bafilomycin A1 negated this effect of UXT. For quantification, similar to the method used for p62 body analysis, we considered pixels with GFP fluorescence intensities exceeding three standard deviations of the mean of whole-cell intensity as regions of SOD1(A4V) aggregates (Fig. 4d), and calculated an aggregation index by summing up the fluorescence intensity of pixels in the regions of aggregates and divided it by the mean GFP fluorescence intensity of the whole cell. Our quantification also showed that the aggregation index gradually increased in control cells, but not in UXT-transfected cells, and that bafilomycin A1 significantly suppressed the effect of UXT (Fig. 4e). Similarly, biochemical analysis of SOD1(A4V) aggregates at different time points showed UXT-dependent disappearance of the aggregates, while bafilomycin A1 suppressed the effect of UXT expression (Supplementary Fig. 3e, f). In addition, siRNA-mediated knocking down endogenous UXT increased the SOD(A4V) aggregates (Supplementary Fig. 3g). Expression of another chaperone, HSP70, as a control did not reduce the aggregates as efficiently as UXT (Supplementary Fig. 3h), and inhibition of its activity by a potent HSP70 family inhibitor, VER-155008, did not block the action of UXT (Supplementary Fig. 3i). Taken together, these results show that UXT mediates the clearance of protein aggregates by autophagy, although we do not exclude the possibility that UXT can also act as a chaperone refolding misfolded proteins in a context-dependent manner.

**UXT binding to the p62 LB domain enhances p62 body formation**. We next investigated how UXT might enhance autophagy. In HeLa/p62KO cells, SOD1(A4V) aggregates cannot be formed (Supplementary Fig. 4a), whereas transfection of WT p62 induced SOD1(A4V)-positive p62 body formation as evidenced by both ubiquitin antibody staining and GFP fluorescence from SOD1 (A4V) (Fig. 5a). The mutant p62(ΔLB) with disrupted UXT binding (Fig. 1h) exhibited significantly reduced SOD1(A4V)-positive p62 body formation when transiently transfected (Fig. 5a, b), as well as stably introduced (Supplementary Figs. 2e, f and 4b, c) into HeLa/p62 KO cells, suggesting that LB domain removal may affect intramolecular and/or intermolecular interactions for p62 body formation. Because the LB domain was identified as the UXT-binding site, we assumed that UXT–p62 interaction might be one of such interactions needed for efficient p62 body formation. Indeed, in HEK293T cells, overexpression of UXT also increased the targeting of p62 to the detergent-insoluble aggregates fraction (Fig. 5c). In contrast, there was reduced targeting of p62(ΔLB) to the insoluble fraction, and this was unaffected by UXT co-expression (Fig. 5c, d). This showed that interaction between p62 and UXT enhances the targeting of p62 to protein aggregates.

UXT is similar in structure to the hexameric chaperone, prefoldin, and is thought to form oligomers[46]. As such, we hypothesized that UXT may form oligomers that can crosslink protein aggregates and p62 proteins, thereby promoting efficient p62 body formation. To get a clue on this, we used an analytic SEC column to analyze the elution profile of GFP-UXT in detergent-free HEK293T cell lysates (Fig. 5e). There were two major peaks observed (Fig. 5f). While the first peak near the void column volume, possibly containing UXT binders, was observed

as in the SEC analysis during purification of UXT (Fig. 3f), the second peak between 440 and 158 kDa was specifically observed in this setting only (Fig. 5f). It may be due to the increased stability of the complex appeared as the second peak in this detergent-free condition and/or the selective detection of GFP-UXT fluorescence in the analytic SEC column instead of the nonspecific detection of protein concentration. The peak corresponding to the size of the monomeric form of GFP-UXT (~40 kDa) was negligible (Fig. 5f). Similarly, when UXT expressed in HEK293T cells without any protein tagging was analyzed in a native gel, it was detected as a possibly large aggregates stuck on the well and also at near 250 kDa marker (Supplementary Fig. 4e, empty and filled triangles), which may correspond to the first and second peaks in the analytic SEC column. Addition of SDS to the samples resulted in its predominant detection at its own size (~15 kDa; Supplementary Fig. 4e). Considering the monodispersity of the peak and the band from overexpressed UXT observed near 250 kDa in the native conditions, it can be assumed that UXT may form an oligomer, although the ~250 kDa complex can be also generated by UXT's interaction with other abundant proteins. To determine the importance of UXT's possible oligomerization with respect to p62 function, we transiently expressed an oligomerization-defective p62 mutant, p62(ΔPB1), in HeLa/p62KO cells and studied its effect on p62 body formation. Consistent with previous reports[47], p62(ΔPB1) did not form p62 bodies and was instead diffused throughout the cytoplasm. In contrast, when mCherry-UXT was transiently co-expressed with p62(ΔPB1), the mutant p62 was observed to form clusters of aggregate structures that colocalized with UXT (Fig. 5g). Although the aggregates showed relatively large and irregular shapes presumably due to the absence of the PB1 domain per se and/or the PB1 domain-mediated oligomerization, the complex seems to be degraded by autophagy because these proteins in detergent-insoluble fraction were increased by blocking autophagy (Supplementary Fig. 4f). To minimize possible artifacts from too much modification on those proteins, we also used an UXT with a smaller tag (HA-UXT) and an oligomerization-defective p62 point mutant, p62(K7A/D69A)[18]. In the experiment as well, we observed the similar UXT-dependent oligomerization of p62(K7A/D69A) (Supplementary Fig. 4g, h). These results suggest that UXT may exist as oligomer that can enhance p62 body formation through direct interaction with p62 and increase its avidity toward protein aggregates.

**UXT binding to p62 is essential for its function with respect to clearance of protein aggregates**. To further validate the importance of the interaction between UXT and p62, we sought to specifically disrupt this interaction. Previous reports have shown that a proline substitution at Leu50 and Leu59 residues that are highly conserved in vertebrates (Supplementary Fig. 5a) resulted in the loss of additional UXT functions, such as regulating notch signaling[48] or as a centrosome component[46]. We thus adapted the double L50P/L59P mutation and tested the effect of the mutation on UXT functions, such as p62 binding (Fig. 1e), p62 clustering by p62(ΔPB1) (Fig. 5g), aggregates clearance (Fig. 4b), and SEC profile (Fig. 5f). As the cyclic side chain of proline usually induces a relatively large structural change, we also generated a UXT mutant that contained alanine substitutions at the leucine residues, UXT(L50A/L59A). Co-immunoprecipitation assays showed that both mutants had significantly decreased p62 binding (Fig. 6a), although the L50P/L59P mutation is more severe than the L50A/L59A mutation (Supplementary Fig. 5b, c). Consistently, both mutants showed defects in enhancing p62 body formation of the clustering-defective mutant p62(ΔPB1) (Fig. 6b, c, blue). Interestingly, we also found that UXT(L50A/

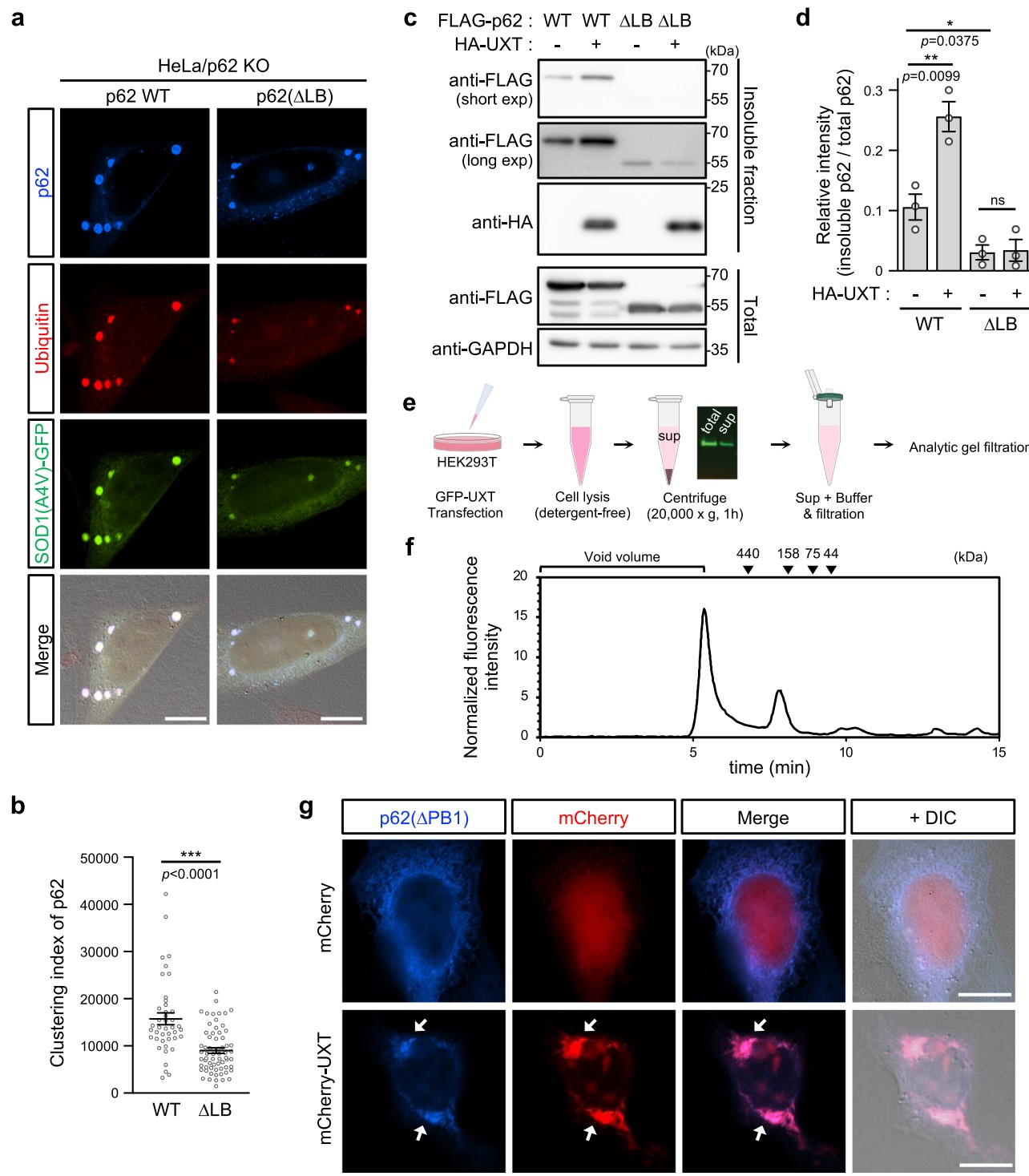

L59A) showed punctate staining similar to the WT, but UXT (L50P/L59P) was completely diffused throughout the cytoplasm (Fig. 6, c, red), suggesting that the L50A/L59A mutant but not the L50P/L50P mutant retains the aggregate-targeting function in spite of the defects in p62 binding. We also found that both mutants failed to decrease the formation of detergent-insoluble SOD1(A4V) aggregates (Fig. 6d, top panel), thereby highlighting the importance of p62 binding in this process. Consistent with our imaging results, targeting of L50A/L59A mutant to the detergent-insoluble fraction was essentially same with WT UXT (Fig. 6d, middle panel), suggesting its aggregate-targeting ability is independent of p62 binding. However, no L50P/L59P mutant was observed in the detergent-insoluble fraction. Importantly, differences were also observed in their SEC profiles. L50A/L59A showed an elution profile identical to that of the WT (compare Figs. 5f and 6f, analyzed simultaneously), but L50P/L59P showed clearly different pattern from WT (Fig. 6f), suggesting that its complex-forming and/or oligomerization activity may be largely reduced. Together, these results show that even with normal aggregate-targeting and oligomerization function, defects in p62 binding observed with the L50A/L59A mutant can reduce p62 body formation and clearance of SOD1(A4V) aggregates. These results also show that the L50A/L59A, but not the L50P/L59P mutation can be used to specifically disrupt the interaction of

**Fig. 5 UXT induces p62 clustering. a** HeLa/p62KO cells were co-transfected with SOD1(A4V)-GFP and either FLAG-tagged p62 wild type or a p62(ΔLB) mutant defective for UXT binding. After transfection, cells were treated with 5 μM MG132 for 6 h to induce p62 body formation and were stained with anti-p62 and anti-ubiquitin antibodies followed by anti-mouse IgG-Alexa-350 and anti-rabbit IgG-TRITC. Representative images are shown. Scale bar, 10 μm. **b** Clustering indexes in cells expressing wild-type p62 or p62(ΔLB) mutant were analyzed and are shown as scatter plots ($n = 42$ and 65 cells examined over three independent experiments, mean ± SEM). ***$p < 0.001$ (two-sided Mann–Whitney test). **c** HEK293T cells were transiently co-transfected with FLAG-tagged p62 (wild type or ΔLB mutant) and HA-UXT or vector control. The detergent-insoluble fraction of cell lysate was analyzed for the presence of these proteins by western blotting. **d** The degree of p62 wild type and p62(ΔLB) mutant detected in the detergent-insoluble fraction was normalized against the degree of their expression (whole-cell lysates) in each condition, and are shown as bar graphs ($n = 3$, mean ± SEM). *$p < 0.05$; **$p < 0.01$; ns not significant (two-sided unpaired $t$ test). **e** To detect an oligomeric form of UXT, detergent-free cell lysates of HEK293T cells expressing GFP-UXT were prepared and their components were separated using a size-exclusion column (SEC) with a fluorometer. **f** SEC profile of GFP-UXT represented by GFP fluorescence of eluant. Molecular weights of the marker are indicated. **g** HeLa/p62KO cells were transiently co-transfected with FLAG-p62(ΔPB1) mutant, the oligomerization-defective mutant, and mCherry-UXT or mCherry-only control. The cells were stained with anti-p62 followed by anti-mouse IgG-Alexa-350, and representative images are shown. p62(ΔPB1) clusters induced by UXT are indicated with white arrows. Scale bar, 10 μm.

UXT with p62, without affecting its aggregate-targeting and oligomerization abilities.

**UXT prevents SOD1(A4V)-induced progressive loss of motor function in *Xenopus*.** The results above suggest that UXT protects the cells from toxic protein aggregates. SOD1(A4V) and SOD1(G93A) are the common toxic gain-of-function mutations commonly associated with in the familial form of ALS, which is clinically characterized by the progressive loss of motor neurons and their function[45]. Expression of mutant human SOD1 in animals also induces the formation of protein aggregates and degeneration of motor neurons, and is widely used as a model of ALS[7,43,49,50]. As *Xenopus tropicalis* develops rapidly and expresses evolutionarily conserved UXT, SOD1, and p62 proteins (Supplementary Figs. 5a and 6a, b), we used *X. tropicalis* as a model to determine if the interaction between UXT and p62 inhibits the formation of toxic aggregates by pathogenic SOD1 mutants. First, we expressed GFP-tagged mutant SOD1 proteins, SOD1(A4V) and SOD1(G93A), in the central nervous system by targeted microinjection of in vitro-transcribed capped RNA (Fig. 7a), and used the touch-induced escape responses, normal, reduced, and no responses, as readouts of progressive loss of motor ability (Fig. 7b and Supplementary Movie 3). Tadpoles expressing SOD1 mutant genes did not show any apparent signs of developmental defects and appeared normal (Fig. 7c). At stage 28, the early swimming state when muscle contract begins to be coordinated, all tadpoles in all groups rapidly swam to the wall of the well and circled around the circumference within second (Fig. 7d). We performed the same test with the same animals injected with SOD1(A4V) 2 days later, at stage 43, to observe any possible deterioration in motor function with time (Supplementary Movie 4). In contrast to the control tadpoles, which showed the same rapid escape responses, most SOD1(A4V)-expressing tadpoles could not swim to the wall (reduced response, 16 out of 34) or could not swim at all (no response, 8 out of 34; Fig. 7e). As the responses of the same tadpoles at stage 28 and at stage 43 were recorded, we concluded that expression of pathogenic mutant SOD1 results in an age-dependent loss of fully developed motor function in *Xenopus*. All tadpoles showed normal function of the cardiac muscle, as such, this defect occurs first in skeletal muscle (Supplementary Movie 5).

Then, we asked whether co-expressing UXT ameliorates SOD1(A4V)-induced progressive loss of motor function. Before the experiment, we confirmed the interaction between *Xenopus* p62 and human UXT by reciprocal co-immunoprecipitation assays (Supplementary Fig. 7a). Expression of human UXT in *Xenopus* almost completely ameliorated SOD1(A4V)-induced loss of motor function (Fig. 7d, e). Importantly, the L50A/L59A mutant, which cannot bind to p62, did not exhibit significant protective effects (Fig. 7d, e), indicating that UXT prevents SOD1(A4V)-

induced loss of motor function through its interaction with p62. Similarly, SOD1(G93A) expression produced a similar but milder dysfunction of motor neurons and UXT suppressed the defect (Fig. 7f, g). To further confirm the involvement of p62 in the UXT-dependent protective effect, we microinjected a translation-blocking antisense morpholino to inhibit the expression of the endogenous p62. Blocking p62 expression did not induce any morphological change (Supplementary Fig. 7b) nor significant defect in motor function at stage 43 (Supplementary Fig. 7c), when the most of behavioral assays in this study were performed, although tadpoles gradually displayed compromised motor ability at stages 44 and 45 (Fig. 7h). However, knocking down p62 almost completely blocked the UXT's protective effect on SOD1 (A4V)-induced toxicity (Fig. 7I, j). These results suggest that overexpression of human UXT may promote the clearance of toxic SOD1(A4V) aggregates by autophagy in a p62-dependent manner, thereby preventing loss of motor function. To test this hypothesis, we examined the subcellular localization of SOD1 (A4V)-GFP in the spinal motor neurons of presymptomatic (i.e., stage 28) and symptomatic (i.e., stage 43) tadpoles (Fig. 7k). Indeed, SOD1(A4V)-GFP was diffused throughout the cell at stage 28, when no motor defects were observed, and had aggregated at stage 43, when deterioration in motor function was apparent. In addition, consistent with this hypothesis, expression of UXT, but not UXT(L50A/L59A), inhibited the accumulation of SOD1(A4V)-GFP aggregates completely at stage 43 (0 cells with aggregates out of 38 transfected cells; Fig. 7k).

**Loss of UXT deteriorates proteotoxicity in *Xenopus*.** Next, we focused on the role of endogenous UXT in *Xenopus*. We first confirmed the existence of endogenous UXT mRNA in whole tadpole and spinal cord tissue at stages 30 and 43, using RT-PCR (Fig. 8a) and in situ hybridization (Fig. 8b). Injection of UXT morpholino into four blastomeres at eight cells stage caused a dose-dependent defect in gross morphology, including a short and bent trunk, a curly tail, a smaller yolk sac, and less pigment (Fig. 8c), and those defects were recovered by co-injection of the morpholino-resistant UXT mutant (Supplementary Fig. 7d), confirming specificity of this phenotype. At the dose of 0.5 ng per blastomere, roughly 40% of tadpoles showed no apparent morphological defects (Fig. 8c), and all these tadpoles gained and maintained full motor function. We asked whether these normally developed UXT hypomorphic tadpoles may have a masked phenotype, such as inability to cope with proteotoxic challenges. Expressing SOD1 mutants, A4V or G93A, which results in age-dependent degeneration of motor function in WT tadpoles, was lethal to these normal-looking tadpoles where endogenous UXT expression was partially knocked down, suggesting that endogenous UXT protein is required to overcome proteotoxicity. To test this idea further, we introduced a manageable proteotoxic

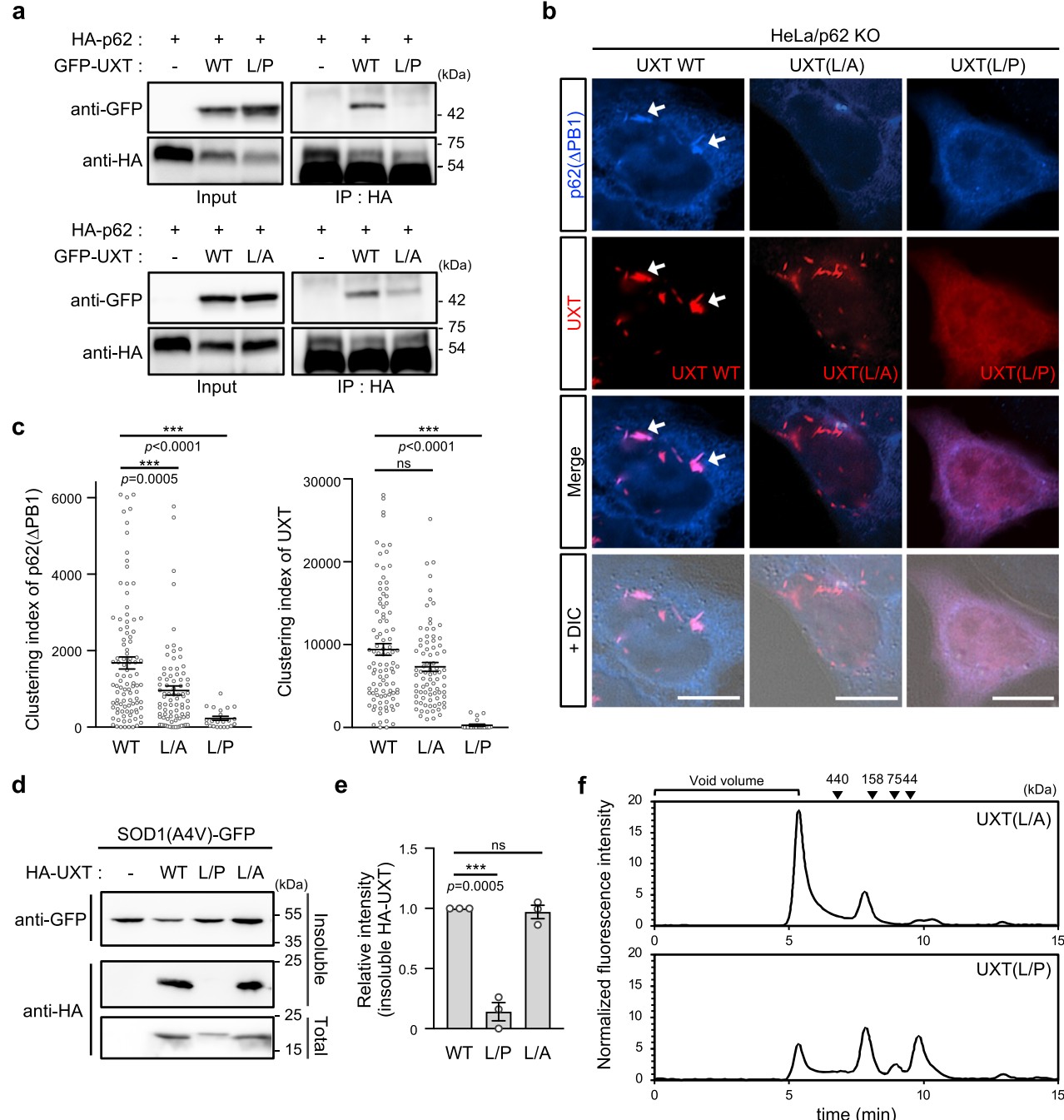

**Fig. 6 L50A/L59A mutations in UXT specifically disrupt interaction between p62 and UXT. a** Interactions between p62 and UXT(L50P/L59P), represented as L/P (upper panel) and p62 and UXT(L50A/L59A), represented as L/A (lower panel) were analyzed. **b** HeLa/p62KO cells were co-transfected with oligomerization-defective p62(ΔPB1) and either UXT wild type, UXT(L50A/L59A), or UXT(L50P/L59P) mutants, and were stained with anti-p62 antibody followed by anti-mouse IgG-Alexa-350. Representative images are shown. p62(ΔPB1) clusters induced by UXT are indicated with white arrows. Scale bar, 10 μm. **c** Clustering indexes of p62 (left) and UXT (right) were analyzed under each condition, and are shown as scatter plots (n = 98, 82, and 22 cells examined over three independent experiment, mean ± SEM). ***p < 0.001; ns not significant (two-sided Mann–Whitney test). **d** The effects of L50P/L59P and L50A/L59A UXT mutations on the amount of detergent-insoluble SOD1(A4V) aggregates. **e** The degree of HA-UXT mutants detected in the detergent-insoluble faction was normalized against that of HA-UXT wild type and shown as bar graphs (n = 3, mean ± SEM). ***p < 0.001; ns not significant (one-way ANOVA using Bonferroni's multiple comparison test). **f** SEC profiles of GFP-UXT(L50A/L59A) and GFP-UXT(L50P/L59P) as detected by GFP fluorescence in the eluant.

stress by overexpressing the WT SOD1 protein, which has little effect in motor function development and maintenance in WT tadpoles (Supplementary Fig. 7e). Indeed, overexpressing WT SOD1 protein in the UXT hypomorphic tadpoles caused motor function deterioration in almost all animals tested, in sharp contrast to the WT tadpoles (Fig. 8d, e). In conclusion, our results show that UXT, through its interaction with p62, reduces the formation of toxic mutant SOD1 protein aggregates, thereby preventing the progressive loss of motor function in *Xenopus*.

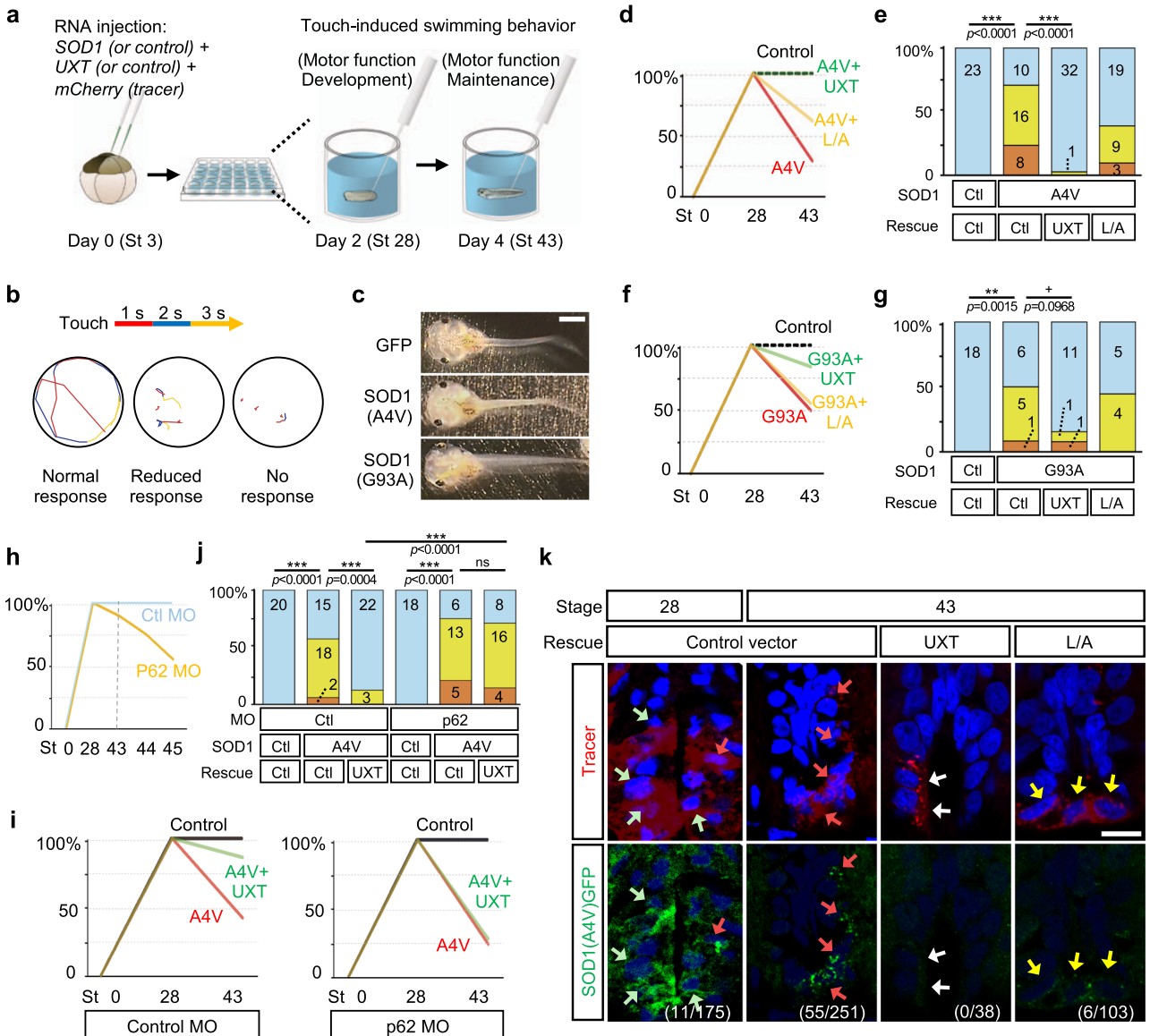

**Fig. 7 UXT prevents SOD1(A4V)-induced progressive loss of motor ability in Xenopus. a** SOD1 and/or UXT in vitro-transcribed capped RNA were expressed in the central nervous system by targeted microinjection. Motor function was assessed by touch-induced swimming responses at stages 28 and 4. **b** Each tadpole was touched three times on its tail. Each trace in a circle indicates swimming trajectory of the same tadpole within 3 s (time after the touch is color-coded). The best response of three trials was used to classify the swimming ability. **c** Injecting SOD1 mutant genes did not result in distinguishable morphological changes. Scale bar, 2 mm. **d** Swimming responses of tadpoles injected with SOD1 mutant and/or UXT (wild type or L50A/L59A mutant) in vitro-transcribed capped RNA are shown. The behavior of tadpoles was recorded in three independent experiments. The Y-axis indicates the percentage of tadpoles showing a normal swimming response, and the X-axis indicates developmental stages. **e** Quantification of **d** at stage 43 with the number of tadpoles in each category is shown. Cyan, yellow, and orange bars represent normal, reduced, and no swimming responses, respectively. The Y-axis indicates the percentage of tadpoles. Ctl control; ***$p < 0.001$ (two-sided Fisher's exact test). **f**, **g** Effect of UXT expression on SOD1(G93A)-GFP-induced motor deficit was tested as in **d** and **e**. $+p = 0.0968$; **$p < 0.01$ (two-sided Fisher's exact test). **h**, **i** Effect of a p62 translation-blocking antisense morpholino (MO) on swimming responses was analyzed as in **d** and **e**. **j** Quantification of **i** at stage 43 as in **e**. ***$p < 0.001$ (two-sided Fisher's exact tests). ns not significant. **k** SOD1(A4V)-GFP was visualized by immunohistochemistry in cell bodies of lower motor neurons localized in the ventral horn of the spinal cord. Red and green arrows indicate SOD1(A4V)-GFP aggregation and SOD1(A4V)-GFP, respectively. Co-injecting UXT completely prevents SOD1(A4V)-GFP aggregation indicated by white arrows. In cells co-injected with UXT(L50A/L59A (L/A)), the SOD1(A4V)-GFP aggregation is often observed as indicated by yellow arrows. Numbers in parenthesis indicate the number of cells with aggregation/the number of transfected cells analyzed. Scale bar, 10 μm.

## Discussion

The process of selective degradation of protein aggregates by autophagy, aggrephagy, requires so-called autophagy receptors, such as p62, NBR1 (ref. [51]), and optineurin[52], working as a molecular bridge between ubiquitinated substrates and LC3/GABARAP family proteins anchoring on the phagophore[53].

Their function can be further enhanced by adaptor proteins that link the aggregates to core autophagic machinery[54]. ALFY, a known autophagy adaptor, binds to p62, ATG5, and phosphatidylinositol 3-phosphate (PI3P) on the phagophore membrane[55,56]. Because ATG5 is the key component for conjugation of PE with LC3 (ref. [57]), ALFY can recruit p62 bodies

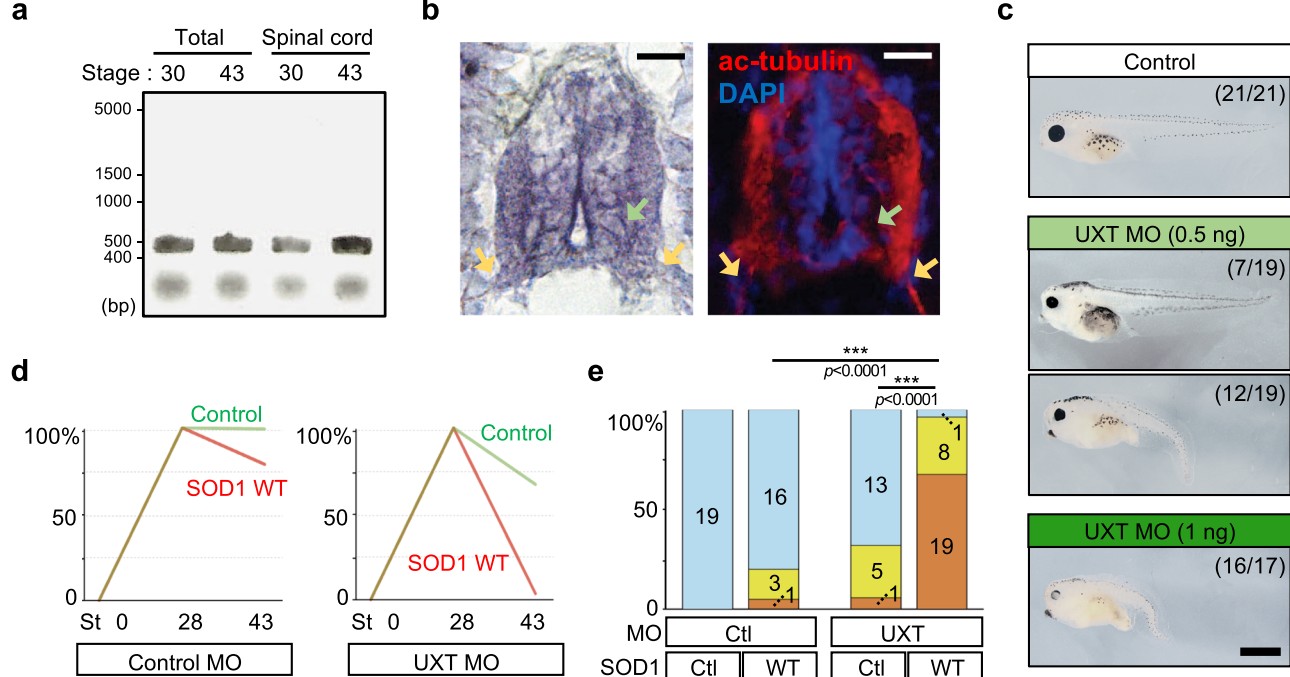

**Fig. 8 UXT is required in coping with proteotoxicity in Xenopus. a** Reverse transcription-PCR of UXT mRNA in whole tadpole and spinal cord tissue. The entire coding sequence of UXT mRNA was amplified with a band of 460 bp. **b** In situ hybridization of UXT mRNA (left) and immunohistochemistry of acetylated tubulin (right). The green arrow indicates a lower motor neuronal cell body and the yellow arrows indicate the axons of these neurons. Scale bar, 10 μm. **c** UXT morpholino (MO) was injected to four blastomeres at the eight cells stage. The injected tadpoles showed a dose-dependent defect in gross morphology in contrast to the control group. The numbers indicate the number of embryos with the same morphology as the representative image out of all examined embryos. Scale bar, 1 mm. **d** Tadpoles injected with 0.5 ng of control or UXT MO together with wild-type SOD1, but showing normal morphology were analyzed for their motor ability as in Fig. 7d. In three independent experiments, each with the full set of groups, the behaviors of 19, 20, 19, and 28 tadpoles were recorded for control, SOD1 WT, UXT MO, and UXT MO + SOD1 WT group, respectively. St stage. **e** Quantification of **d** at stage 43. Cyan, yellow, and orange bars indicate normal, reduced, and no swimming responses, respectively. Ctl control. ***p < 0.001 (two-sided Fisher's exact tests).

near the PI3P-containing phagophore where it induces the formation of PE-conjugated LC3, thus facilitating the p62-dependent autophagy. In our attempt to find such autophagy adaptors, we identified UXT as a p62 binder and demonstrated that their interaction can prevent proteotoxicity by facilitating autophagic degradation of protein aggregates.

In accordance with our hypothesis that autophagic clearance of toxic aggregates involving UXT–p62 interaction is required for maintenance of motor neuronal health and function, missense mutations in p62 are indeed linked to ALS in human[58], and p62 loss-of-function exacerbates motor neuron degeneration in ALS models in mouse[59] and zebrafish[60]. In case of UXT, there are no known mutations associated with neurodegenerative diseases, presumably because functionally harmful mutations in UXT may be lethal as in the *Xenopus* study, where its knockdown leads to severe developmental defects (Fig. 8c and Supplementary Fig. 7d). Intriguingly, both UXT[61] and p62 (ref. [59]) are known to interact with als2 protein, associated with ALS in human.

Then, how does UXT enhance the autophagy function of p62? p62 is a multi-domain protein that functions as a receptor that links protein aggregates to the autophagy machinery[15]. The N-terminal PB1 domain mediates homo-oligomerization of p62, which can increase its affinity for protein aggregates and/or the phagophores[15,18,19], is also crucial for p62 function. Indeed, deletions or mutations in the PB1 domain reduce LC3 binding and p62 body formation[18,62], illustrating the importance of higher-order oligomer formation of p62 with respect to its function in autophagy. Similarly, the oligomerization of an autophagy receptor in yeast, Cue5, is also known to play a role in

targeting the protein aggregates for autophagy[63]. Structural studies have shown that the PB1 domain forms long filamentous helical polymers though successive electrostatic interactions[19]. As the polymerization mainly depends on interactions between the acidic and basic surfaces on the two opposite faces of PB1 domain in a head-to-tail manner, the polymerization leads to a linear polymer without branching. Moreover, the linear p62 polymer forms a helical structure with PB1 domains closely packed inside of the helix[19]. In this highly packed helical structure of p62 polymer, bending of the polymers would not be favored due to the steric hindrance from each molecule in the structure, thus limiting its degree of freedom and efficiency of binding to protein aggregates and LC3. As UXT is expected to form an oligomer that can provide multiple binding sites for p62, interactions between the p62 filament and oligomeric UXT may possibly circumvent this limitation by weaving the filamentous p62 polymers to create a reticulated p62 structure that can serve as a high-affinity trap for ubiquitinated protein aggregates and LC3 (Supplementary Fig. 8b). This may explain the ability of UXT to enhance the function of p62 in autophagy.

Second, the chaperone function of UXT may also enhance p62-dependent autophagy through the ubiquitin-independent recruitment of p62 to protein aggregates. UXT belongs to a prefoldin α-like family that along with the prefoldin β family, forms a hetero-hexameric jellyfish-like molecular chaperone complex[31,64]. If UXT can form a similar oligomeric structure as prefoldin, the internal surface of the UXT in the structure would interact with unfolded protein (Supplementary Fig. 8c). We observed that UXT can be localized in the detergent-insoluble

fraction normally even when p62 binding is impaired by the L50A/L59A mutation (Fig. 6d), showing that UXT can interact with unfolded proteins and recruit p62 to aggregates independent of its ability to interact with ubiquitinated target proteins. In fact, several ubiquitin-independent- and chaperone-mediated selective autophagy (CMA) mechanisms, e.g., aided by the aggregates targeted heat-shock protein and its co-chaperone BAG3, have been identified[65,66]. A distinct feature of CMA is the direct delivery of misfolded proteins by chaperones to the lysosome, where specific lysosomal membrane proteins translocate the misfolded protein into the lysosomal lumen without the involvement of phagophores[67]. As UXT colocalizes with LC3, a marker of the growing phagophore, and as its protective function is dependent on the interaction with p62 (Fig. 7i), the action of UXT is distinct from that of CMA and appears to involve the aggrephagy system. This suggests a role for chaperones in aggrephagy.

In conclusion, we have identified UXT as an autophagy adaptor that cooperates with p62 to enhance the removal of protein aggregates by autophagy. Because increasing UXT protein level could relieve the proteotoxicity (Figs. 2d and 7d, e), we believe that our findings may provide a therapeutic strategy for protein aggregation-mediated diseases, e.g., by a gene delivery of UXT or by a selective inhibition of UPS-dependent degradation of UXT (Fig. 3a). This study also creates a paradigm for cooperation by chaperones with the aggrephagy system to clear misfolded proteins.

## Methods

**Cell culture**. HEK293T (ATCC), HeLa cells (ATCC), HeLa cells expressing GFP-CFTR(ΔF508)[37], p62 knockout HeLa cells (HeLa/p62KO; generated by CRISPR-Cas9 system), HeLa cells expressing GFP-LC3 (HeLa/GFP-LC3; generated by lentiviral transduction), and HeLa/p62KO cells expressing p62 WT or mutants (generated by lentiviral transduction of GFP-tagged p62 constructs into HeLa/p62KO cells) were maintained in Dulbecco's modified Eagle's medium (HyClone) supplemented with 10% (v/v) fetal bovine serum (Gibco), and 1% penicillin–streptomycin (HyClone) at 37 °C with 5% CO$_2$ in an incubator. For the starvation experiment, cells were incubated in EBSS (HyClone). DRG neurons were isolated from E13.5 mouse embryos and cultured in neurobasal medium (Thermo Fisher) supplemented with 1% penicillin–streptomycin (HyClone), 5-fluoro-2-deoxyuridine (final concentration, 2.5 μg/mL; Sigma) and uridine (Sigma), 1× B27 supplement (Thermo Fisher), GlutaMAX supplement (Thermo Fisher), and Nerve Growth Factor 2.5 s (Life Technologies, 100 ng/mL) at 37 °C in an atmosphere containing 5% CO$_2$ in an incubator.

**Plasmids and cloning**. pcDNA3-HA-p62 was previously described[68]. pEGFP-n3-SOD1 WT and mutants (A4V and G93A) were provided by S.M. Kang. pOTB7-UXT clone (hMU006749) was obtained from the Korea Human Gene Bank, Medical Genomics Research center, KRIBB, Korea. Plasmid constructs were generated using restriction enzyme digestion and cloning, and mutant constructs of p62 or UXT were prepared by site-directed mutagenesis. HA-Xenopus p62 was purchased from GenScript. The constructs, cloning methods, and sequences of primers used are described in the Supplementary Data 1.

**Antibodies**. The following antibodies were used for immunoblotting (IB), immunoprecipitation (ICC), immunoprecipitation (IP), and immunohistochemistry (IHC): mouse monoclonal anti-ubiquitin (1:200 (ICC), Abcam, ab7254), rabbit monoclonal anti-ubiquitin (1:200 (ICC), Abcam, ab134953), mouse monoclonal anti-p62 (1:2000 (WB), 1:500 (ICC), Abcam, ab56416), rabbit monoclonal anti-p62 (1:2000 (WB), 1:500 (ICC), Abcam, ab109012), rabbit polyclonal anti-p62 (2 μg (IP), Abcam, ab101266), mouse polyclonal anti-UXT (1:500 (IB), 1:200 (ICC), Abcam, ab168678), rabbit polyclonal anti-UXT (1:1000 (IB), Sigma, AV51213), rabbit monoclonal anti-LC3A/B (1:1000 (IB), 1:200 (ICC), CST, 12741), mouse monoclonal anti-GAPDH (1:2500 (IB), Santa Cruz, sc-32233), rabbit polyclonal anti-cleaved Caspase-3 (Asp175) (1:300 (ICC), CST, 9661), rabbit polyclonal anti-acetylated alpha Tubulin (acetyl K40) (1:300 (IHC), Abcam, ab12536), mouse monoclonal anti-GFP (1:2500 (IB), 2 μg (IP), Santa Cruz, sc-9996), mouse monoclonal anti-FLAG (1:2500 (IB), Sigma, F1804), rabbit polyclonal anti-HA (1:2000 (IB), 2 μg (IP), Thermo Fisher, 71-5500), normal mouse IgG (1:500 (ICC), Santa Cruz, sc-2025) and normal rabbit IgG (1:500 (ICC), 2 μg (IP), CST, 2729), goat anti-mouse-HRP conjugated (1:10000 (IB), Thermo Fisher, 31430), goat anti-rabbit-HRP conjugated (1:10000 (IB), Thermo Fisher, 31460), anti-rabbit IgG-Alexa Fluor 350 (1:1000 (ICC), Thermo Fisher, A11046), anti-

mouse IgG-Alexa Fluor 350 (1:1000 (ICC), Thermo Fisher, A11068), anti-rabbit IgG-Alexa 555 antibody (1:1000 (IHC), Thermo Fisher, A31572), anti-rabbit IgG-TRITC (1:1000 (ICC), Jackson Laboratory, 711-025-152), and anti-mouse IgG-TRITC (1:1000 (ICC), Jackson Laboratory, 715-025-151).

**Yeast two-hybrid screening**. Yeast two-hybrid screening of p62 was performed using the yeast strain PBN204 that contained three reporters (URA, lacZ and ADE2). Amino acids 114−224 of p62 were cloned as an EcoRI/XhoI fragment into EcoRI/SalI sites of a pGBKT vector. The pGBKT plasmid contained the cDNA encoding the DNA-binding domain of GAL4 (GAL4 BD), a *TRP1* selection marker, *Kan*$^R$ marker and the replication origin sequence of yeast 2 μ plasmid. The junction between GAL4 BD and the p62 was verified by DNA sequencing. Using a cDNA library, human thymus cDNA inserts were cloned as EcoRI/XhoI fragments into a pACT2-containing the cDNA encoding GAL4-activating domain. Yeast transformants of the p62 bait and human thymus cDNA AD library were plated on selection medium-leucine, tryptophan, and uracil. After selecting yeast colonies on uracil-deficient media, the beta-galactosidase activity was monitored. The screened URA$^+$ and lacZ$^+$ colonies were then tested for growth on adenosine-deficient media. For this, three independent reporters with different types of GAL4-binding sites were utilized to reduce the number of false positive candidates. To further confirm the interaction, DNA isolated from the prey URA$^+$ADE$^+$LacZ$^+$ candidates was amplified by PCR or *Escherichia coli* transformation. The amplified candidate prey DNA samples were reintroduced into yeast with the p62 bait plasmid or with a negative control (vector) plasmid, and then the growth of yeast transformants was studied.

**p62 knockout using CRISPR/Cas9**. p62 was knocked out in HeLa cells using the GeCKO (Genome-scale CRISPR Knock-Out) system[69] with two target sequences, p62KO-1 (5′-CGGGTCCGGGACCCTGCGAGCGG-3′) and p62KO-2 (5′-TAGTG CGCCTGGAAGCCGCCAGG-3′). CRISPR/Cas9-resistant p62 constructs containing the silencing mutation in the 3′ three nucleotides of the target sequences (PAM sequence) were used for p62 expression in HeLa/p62KO cells.

**Co-immunoprecipitation**. Polyethylenimine was used to transiently transfect HEK293T cells, and the cells and constructs were incubated at 37 °C for 24 h. Cells were lysed with lysis buffer containing 20 mM HEPES, pH 7.4; 150 mM NaCl; 1% NP-40; 1 mM EDTA; and protease inhibitor cocktail (Roche). After centrifugation for 20 min at 17,000 × g, the resulting supernatants were incubated with antibodies for 4 h at 4 °C, following which they were incubated overnight with protein A or G agarose beads (Pierce) at 4 °C. The pellets were washed four times with lysis buffer and proteins were analyzed by IB.

**Quantitative RT-PCR**. Total RNA was extracted using TRIzol (Thermo), and mRNA was reverse transcribed into cDNA using the PrimeScript RT reagent kit (Takara). Quantitative RT-PCR was performed using a LightCycler 2.0 (Roche), with a LightCycler 480 SYBR Green I Master kit (Roche) and specific primers. The sequences of primers used are listed in the Supplementary Data 1. Expression levels of UXT were normalized to those of actin mRNA. The relative amount of mRNA expression was calculated by using the ΔΔCt method.

**Preparation of the detergent-insoluble fraction**. Cells were lysed with lysis buffer A containing 20 mM HEPES, pH 7.4; 150 mM NaCl; 1% Triton X-100; 1 mM EDTA; and protease inhibitor cocktail (Roche). Five percent of the resulting lysate was saved as total protein. After centrifugation at 17,000 × g (4 °C for 20 min) the supernatant, or the detergent-soluble fraction, was removed. The pellet, or the detergent-insoluble fraction, was washed with lysis buffer A and solubilized with lysis buffer B containing 20 mM HEPES, pH 7.4; 150 mM NaCl; 1% Triton X-100; 1% SDS; 1 mM EDTA; and protease inhibitor (Roche). A total of 15–20 μg of proteins were loaded per sample. Uncropped scans of blots are supplied in the Source data file.

**Immunocytochemistry**. Lipofectamine 2000 (Invitrogen) was used to transfect cells cultured on gelatin-coated coverslips with various constructs. After transfection, cells were incubated for 24 h, following which, they were fixed using 3.7% formaldehyde, and then permeabilized with phosphate-buffered saline (PBS) containing 0.1% Triton X-100 (PBST). Cells were treated with the blocking solution (1% BSA, 22.52 mg/mL glycine, and 0.1% gelatin in PBST), and were stained first with respective primary antibodies and subsequently with fluorophore-conjugated secondary antibodies diluted in blocking solution. Coverslips were mounted on microscope slides with fluorescence mounting medium (Dako) with or without Hoechst (Invitrogen). Images were acquired using a fluorescence microscope (Ti-E, Nikon) equipped with a 100× (1.4 N.A) Plan-Apochromat objective lens and a charge-coupled camera device (DS-Qi2, Nikon). The images were processed using NIS-Elements AR (Nikon) and/or ImageJ (NIH) software. A PLA assay was performed using a Duolink in situ red starter kit mouse/rabbit (Sigma-Aldrich, DUO92101), according to manufacturer's instructions.

***Xenopus* experiments**. *X. tropicalis* embryos were generated by in vitro fertilization, raised in 0.1× Modified Barth's Saline at 21–24 °C, and staged according to the tables provided in Nieuwkoop and Faber (1967) and Xenbase (http://www.xenbase.org/). The mMESSAGE mMACHINE kit (Ambion) was used to transcribe the capped RNAs in vitro. Translation-blocking antisense morpholino oligonucleotides were purchased from Gene Tools, LLC. The sequence information is in Supplementary Data 1. A total of 450 pg (200 pg encoding EGFP-fused SOD1 mutants or control (EGFP), 200 pg encoding UXT or control (mCherry), and 50 pg encoding a tracer (mCherry)) of RNA per blastomere was microinjected into two dorsal blastomeres that give rise to the central nervous system. A total of 0.5 or 1 ng of morpholino oligo was injected, as described in the figure legends. One injected embryo was placed per well in 24-well culture plates. Under a dissection microscope equipped with a video camera, each tadpole was touched at least three times on its tail with the tip of a plastic microloader. Response was classified as a normal response when the tadpole swam to the edge of the well and then around the well within 3 s. Response was classified as a reduced response when the distance that the tadpole swam was shorter than the radius of the well. Response was classified as no response when the tadpole failed to move forward at all. The best score of three consecutive tests was recorded for each tadpole. Tadpoles were fixed overnight in 4% paraformaldehyde prepared in PBS at 4 °C. The fixed embryos were sectioned using a cryostat, and imaged under a LSM confocal microscope (Zeiss) equipped with a 63× C-Apochromat (numerical aperture 1.2) objective, with a *XY* zoom factor of 1.50. All experiments complied with protocols approved by the Yonsei University College of Medicine Institutional Animal Care and Use Committees.

**Statics and reproducibility**. Statistical analysis was performed on data from three or more biologically independent experiments using GraphPad Prism (v 9.0.0, GraphPad Software). Statistical parameters including precise value of *n* (total number of analyzed cells, microscopic fields, independently repeated experiments), *p* values, definitions of error bars with a measure of center, and statistical tests used are indicated in the figures and figure legends. All experiments were done three times, unless otherwise stated.

**Reporting summary**. Further information on research design is available in the Nature Research Reporting Summary linked to this article.

## Data availability
The main data supporting this study are available within this article. All other data supporting the findings of this study are available from the corresponding author upon request. Source data are provided with this paper.

## Code availability
The source code is provided as Supplementary Code 1 and available at https://doi.org/10.5281/zenodo.4563694 (ref. [70]) without restrictions. This work uses MATLAB (The MathWorks Inc).

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

## Acknowledgements

This study was supported by the Basic Science Research Program through the National Research Foundation of Korea (NRF-2019R1A2C2008067, NRF-2015R1A4A1041919, NRF-2017R1A2B4002683, and NRF-2018R1A5A2025079), Samsung Science and Technology Foundation (SSTF-BA1602-13), and a Korea University Grant.

## Author contributions

M.J.Y., H.J. and C. Kim designed the project. M.J.Y. (biochemistry and cell biology), B.C. (*Xenopus* experiment), E.J.K. (biochemistry), J.O. (*Xenopus* experiment), C.Y. (cell biology), Y.-G.C. (biochemistry), and J.L. (cell biology) performed experiments. M.J.Y., H.J., C. Kang, H.K.S., Y.K.K., J.-S.W., Y.C., E.-J. C. and C. Kim analyzed the data. M.J.Y. and C. Kim wrote the paper, which was edited by C. Kang and H.J.

## Competing interests

The authors declare no competing interests.
