## [Peer Review File · Nature Communications]

REVIEWER COMMENTS

Reviewer #1 (Remarks to the Author):

In the manuscript by Yoon, M. J. et al. the authors show that the prefoldin-like protein UXT interacts with the autophagy cargo receptor p62 through p62's LB domain and thereby promotes SOD aggregate removal. UXT co-localizes with aggregated p62 or occasionally LC3 in cellular punctae and assists in the clearance of p62 bodies via autophagy. The authors hypothesize that the hexameric state of UXT helps cross-linking p62 aggregates inside cells as disruption of this oligomeric state reduces aggregate removal. Interestingly, aggregate removal by a p62 truncation lacking the PB1 domain responsible for homo-oligomerization can be rescued by UXT transfections. Finally, Yoon et al. show evidence that UXT overexpression in *Xenopus* tadpoles prevents SOD aggregate accumulation in neurons and thereby prevent loss of motor ability.

This manuscript demonstrates for the first time that the UXT co-chaperone interacts with p62 and is involved in aggrephagy. It covers a broad range of different cellular methods and model systems including cell lines as well as tadpoles. The manuscript has the potential to be published, once the following points are addressed:

Major points:

1. The authors show that UXT transfection mediates the clearance of SOD aggregates (Figure 5d). However, it has also been shown that prefoldin co-assembles with chaperonins such as TriC (PMID: 30955883). The authors make the claim that UXT's activity of aggregate removal is due to autophagy. Nevertheless, chaperones with UXT's co-chaperone activity should also be considered as a contributing factor to dissolve aggregates. For instance, as a control it would be helpful to see the aggregate removal effect mediated by other chaperones such as Hsp70 and TriC. Additionally, does the UXT L/P mutant show that this is a specific effect due to p62 binding and autophagic clearance?
2. In order to demonstrate that UXT is involved in autophagy, the authors essentially use an autophagy flux assay by inhibitor Bafilomycin. Currently, Figure 3a only shows only an increase in UXT levels. Please include p62 as well as LC3 Western blots. LC3I to LC3II conversion alongside p62/UXT increase will make the point clearer and point strongly to autophagy. Quantification of all band intensities will be helpful to verify bona fide autophagy turnover. This type of standard autophagy flux assay is generally recommended by a community consensus paper (Klionsky et al., 2016 Guidelines for the use and interpretation of assays for autophagy).
3. It is notable that very little p62 Δ LB in comparison to wt-p62 can be found in the so-called insoluble fraction, independent of UXT addition (Figure 5c). Please quantify protein amounts in the different lanes, while correcting the varying amounts in loading control. Upon LB domain removal p62 filament formation or p62 interaction with other proteins is already significantly affected. No explanation is offered in the text. Please clarify as this p62 Δ LB mutant is used throughout the manuscript.
4. The discussion section of the manuscript should be heavily reworked:
 - a) First, it is very short and should be extended. There is a whole body of work regarding aggrephagy that could be put into context.
 - b) Second, there is also more known on the mechanisms of prefoldin as a co-chaperone.
 - c) Third, the hypothesis of p62 weaving is not easy to follow and should be highlighted as speculation.
5. The following conclusion is unclear and not justified by the results:
page 7: "If this is the case, the interaction between the hydrophobic surface of UXT and misfolded proteins under pro-aggregation conditions may prevent degradation by UPS, and promote autophagy via the recruitment of p62 to the UXT-aggregate complex." The accumulation of UXT upon MG132 treatment is consistent with UXT removal by the UPS. It is far-fetched to state that the hydrophobic interaction is critical here as it has not been demonstrated by the experiments.

Minor points:

1. Related to major point 1, please include more background information on UXT in the introduction section.

2. It is very easy to miss that additional efforts have been made to identify the critical stretch of the p62 LB binding region. This data is hidden in the Supplementary Figure 4. It should find a more pronounced place close to Figure 1f. Please improve phrasing in the results section and name the relevant residue region to make the vague statement for concrete:

"We subsequently used binding analysis with p62 deletion mutants to further map the UXT binding site to the N-terminal half of the LB domain (Fig. 1f and supplementary Fig. 4)"

3. In Figure 1j, please include a p62 Western blot to verify the presence of p62 in the insoluble fraction.

4. Figure 3c: Structural model of UXT including electrostatic surface plot. Please describe how this model was generated in more detail. Is it based on a homologous structure? If yes, based on which protein, what was the sequence identity and how was this model generated?

5. Figure 3d: The intensity profiles are not showing colocalization conclusively as they are the intensity overlays of one line in one cell, which makes it hard to judge whether this is statistically significant. A better way to depict this is to make a scatterplot, where UXT intensity is plotted against p62 intensity and a correlation coefficient is calculated and compared to p62 + LC3 and UXT + LC3.

6. Figure 3f: The p62 and LC3 bands have very low intensity above background level. Please show a SEC chromatogram and an SDS-PAGE or native PAGE analysis of the different fractions. A ponceau stain of the blot would also be helpful. Presence of a large series of other proteins could indicate additional substrate targeting to the autophagosome.

7. Figure 7d and e. There is no unit of the y-axis. How is this 0 – 100 value obtained?

Reviewer #2 (Remarks to the Author):

UXT is a prefoldin-like chaperone and Yoon et al used a yeast two-hybrid screen to identify UXT as binding to p62/SQSTM1. UXT oligomerizes and binds to p62 to increase the ability of p62 to form p62 bodies and degrade protein aggregates by autophagy. A *Xenopus* model of motor neuron degeneration caused by expression of the ALS SOD1(A4V) mutant is used to show that overexpression of UXT delays motor neuron degeneration. The L50A/L59A mutation of UXT which inhibits its binding to p62 is unable to prevent motor neuron degeneration in the *Xenopus* model.

I find this paper very interesting and bringing the novel feature of an oligomerizing chaperone binding directly to p62 to enhance the ability of p62 to degrade protein aggregates by autophagy. However, as detailed below, the data shown do not always justify the conclusions made and more control experiments are needed as well as better quantifications of some of the data. A major problem is the use of overexpression by transient transfections in some experimental settings when stable cell lines with more physiological relevant expression levels should be used instead.

1. Fig. 1b: p62 is overexpressed (OE) by transient transfections in HeLa cells KO for p62. The OE of p62 appears quite high and it would have been better to use stable cell lines with close to endogenous expression level of WT and deletion mutant of p62.

2. While appreciating the quantification method developed to measure what the authors call clustering index, the authors should also quantify the number and size distribution of the p62 bodies for WT and deltaZZ-LB in Fig 1 to reveal the difference in these parameters.

3. In Fig. 2 data from a cell death assay based on blebbing as a readout is shown. Blebbing structures are compatible also with other types of cell death, as apoptosis. The authors should clarify which kind of cell death is studied here. Analysis of blebbing is not enough and other assays are needed such as i.e. Hoechst staining of nuclear fragmentation, annexin V staining or flow cytometry using propidium iodide.
4. Is p62 required for reduction of cell death upon overexpression of UXT in the model used in Fig. 2?
5. The blot in Fig 3a needs to be quantified. It does not seem very clear from the blot alone that UXT is stabilized by Baf, indicative of UXT being degraded by autophagy while the stabilization by proteasomal inhibition looks very clear. Use of ATG5 or ATG7 KO cells should be done to determine if UXT accumulates when macroautophagy is inhibited.
6. Will UXT accumulate in the absence of p62?
7. Fig 3d needs better quantification of colocalization with statistics.
8. The data in fig4e should be accompanied by western blots of the insoluble fractions at the relevant time points.
9. Figure 5a: Again p62 overexpression levels are too high. Dot size in HeLa cells are much bigger than endogenous p62. These experiments are better performed in stable cell lines.
10. Are the big aggregates (p62 Δ PB1 + UXT-mCherry) in Fig. 5f degraded by autophagy?
11. Fig 6b shows irregular shaped aggregates that do not look like the round p62 bodies. The aggregates depend on UXT interacting with p62 and this suggest they are formed by UXT and not by p62 Δ PB1.
12. How does knock down of p62 affect the swimming ability in the *Xenopus* SOD model?

Minor:

In the Introduction there are references to several rather old review that should be replaced by more updated ones.

The introduction is very short and should contain some background information on prefoldin-like chaperones and UXT.

Is it known if UXT may be lost or mutated in ALS cases?

On page 3 the authors write: "phagophore (the autophagic membrane structure)"

This needs to be corrected to either (the forming autophagosome) or (isolation membrane).

On p4, 5 lines from top: The role of the KIR domain is not unclear as it mediates binding to KEAP1 and this is modulated by phosphorylation in the KIR motif.

Delete "were" in line 5 from top on p 9

Ref 37 is the same as ref 15 only that the journal name is missing

The formal name of p62, SQSTM1 (sequestosome-1) should be given when first mentioning p62.

Actually, this name was given by J. Shin who discovered p62.

Reviewer #3 (Remarks to the Author):

This study identifies UXT in a yeast two-hybrid screen as an interacting partner with the ZZ/LB domains of p62, an aggregate degradation mediator. The paper demonstrates a role for UXT in the degradation of ubiquitinated aggregates via autophagy through its interaction with p62. In addition, the study demonstrates a novel role for UXT which rescues a CNS defect phenotype in *X. tropicalis*.

Major weaknesses:

One weakness of the study is that all of the phenotypic data demonstrating the role of UXT in aggregate degradation relies on UXT over-expression rescue experiments, rather than knockout or knockdown of endogenous UXT. While it is interesting that over-expression of UXT can rescue various findings, including aggregation of proteins and cell death in the presence of MG-132, there should be some evidence of the function of endogenous UXT in these pathways, and not only the endogenous interaction between p62 and UXT shown in Fig. 1g/1h. The manuscript should not be accepted without results showing perturbation of endogenous proteins affect function.

Additionally, the claim that UXT forms a hexamer with a molecular weight of about 200 is somewhat unfounded, since the peak at 200 kDa on the SEC may also be a complex or aggregate of UXT with other proteins.

Minor weaknesses:

1. In Fig. 1f, can the authors speculate as to why the dZZ mutant p62 binds to UXT better than the WT?
2. In Fig. 2c, what is the role of endogenous UXT in the neuronal cells? Is it expressed? Over-expression of UXT results in improved cell survival in the presence of MG-132. Is this due to the presence of more UXT? How much over-expression is required to see this effect?
3. For Fig. 3a the effect of the autophagy inhibitor on UXT levels is very modest. For Fig. 3d what do the dotted arrows show?
4. In Fig. 4a, it still appears that UXT is forming aggregates in the SOD1 WT GFP cells. What can explain this? These clusters do not appear to be that different from the clusters in the SOD1(A4V) cells. Can the authors use their quantification system to show this difference?
5. In Fig. 5e, the authors need more evidence that the ~200 kDa peak is an oligomer of UXT and not also a complex with other proteins. If the >669 kDa may be a complex of UXT associated with other proteins, why is this not true for the other peak? One way to do this would be to show a silver stain of the 200 kDa peak and show that there is only one band at the size of UXT. Another alternative would be to show a non-denaturing gel Western blot with UXT antibody and show a band at this larger size. The data here is not enough evidence to claim that it is a hexamer.
6. Can the authors please specify the expected monomeric size of UXT-GFP for Fig. 5e and Fig. 6e? It is a bit hard to tell which peak is meant to be the "monomer". Since GFP is around 27 kDa, it is also hard to calculate how 200 kDa would be a hexamer.
7. The authors state that the L50A/L59A but not the L50P/L59P mutant of UXT retains some aggregate targeting function in spite of the defect in p62 binding. However, if you look at the co-IP in Fig. 6a, the L50A/L59A mutant does retain some p62 binding. Is it possible that the difference in aggregate targeting is actually due to the total loss of L50P/L59P binding vs. the partial loss in L50A/L59A binding? How do you know that UXT and SOD1(A4V) are both expressed when co-injected?
8. The claim in the discussion that UXT could be part of a therapeutic strategy should be elaborated on – how can you enhance the function of a molecular chaperone therapeutically?

Reviewer #4 (Remarks to the Author):

The work of Yoon et al. aims to provide novel insights into the impact UXT, a prefoldin-type chaperone on the function of p62 in autophagy. The authors claim that the aggregation of proteins is prevented by the physical interaction of UXT with substrate helping p62 to eliminate accumulated protein, which otherwise can confer pathological cytotoxicity. Thus, the manuscript deals with an important and interesting question. In general, the technical quality and the presentation of the data is high. However, the work is not always solid in the present form. Nevertheless, the manuscript is of potential interest for the readership of Nature Communication after revision.

My concerns:

The author should provide the full western blots of their protein analyses, which include a proper size indication. A table of the commercial antibodies including the order numbers, dilution, and crossbench references showing their specificity would be helpful. Further, a table listing all expression constructs and their composition with the expected size of the resulting protein. The RAW data of their statistical analyses should be provided in the supplement. To me, it is not clear how the authors scanned & normalized their protein blots. How much protein has been loaded per sample? For the PCRs, the authors should provide the accession numbers of the transcripts analyzed and whether the primers

target alternative splice forms, or in the case of overexpression can discriminate between the endogenous and exogenous transcripts.

Figure 1:

A tricky issue of the paper is that the identified UXT chaperone offers a sticky surface, which UXT uses to bind to unfolded or misfolded proteins. Though the yeast two-hybrid screen has been done properly in principle, it cannot be excluded that an even minor proportion of the bait (ZZ&LB domain of p62) is not folded correctly and thus forms an artificial binding surface for chaperones as UXT. In many YTH screens or protein IP/Mass spectrometry analyses, chaperons of different kinds depending on the complexity of the library, tissue, cell line are always in the top 10 candidates. Thus, the question of whether the physical interaction of UXT and (ZZ-LB)-p62 is real, needs even more attention. Using co-IP of tagged-versions (HA and GFP) they show that HA-p62 and HA-p62deltaZZ pulled GFP-UXT but not delta-LB. Thus, the LB domain should contain the binding site for UXT. Here, I miss the IP of UXT, showing that p62 is bound in a complementary manner. Moreover, co-IPs of endogenous UXT / p62 is essential, the respective antibodies are listed and used throughout the paper. For better control, the authors should employ their p62-KO cells in PLA, not showing just an IgG control. However, the PLA cannot fully replace the endogenous IP, since it is not surprising to find a chaperone close to protein aggregates with p62. It may simply indicate that both, p62 and UXT, indeed act together to clear protein aggregates, but they must not physically interact necessarily. J. Protein's standard is missing.

Figure 2:

It is an interesting observation that overexpression of (mCherry)-UXT seems to reduce the proportion of dying DRG neurons when aggregates cannot become degraded by the proteasome. WT-UXT is 18kDa and mCherry is 27kDa in size. Thus, the tag is much bigger than the tagged protein any may interfere with UXT activity/function (the big ball covers -even partially- the full activity/function of the small ball). An easy control would be to compare mCherry-UXT with an untagged version or with a much smaller tag (Flag). I wonder whether UXT blocks the progression of the cell cycle in your experiments. Thus, a comparison of the cell cycle state upon UXT treated vs. WT could be interesting.

Figure 3:

The authors present data, which show that p62 and UXT are close to autophagosomes at the same time. Please, indicate the secondary antibodies used in the legend of figure 3 (and in general). The Western blot in f shows three fractions of the elution of a 10His-eYFP-UXT version, the resulting fusion protein has an estimated size of 45kDa but the blot indicates 15kDa. LC3B-II normally runs a little bit faster (smaller) than 15kDa, the authors should check. The author should clarify this discrepancy showing the raw data. Again, the authors should explain why they didn't try to detect the endogenous proteins only (e.g. native IP of UXT followed by mass spectrometry).

Figure 4:

The authors show that inhibition of autophagy rescues UXT induce degradation of mutant SOD1 aggregates. In figure 4a SOD1-WT also forms clusters (weak green signals), which are of the same pattern as mCherry-UXT. Is this an artifact of overexpressed SOD1? Shouldn't we expect SOD1-WT protein in mitochondria? In the text (page 9) it must-read ... "using a live cell imaging system (Fig. 2d)" not 2b. The authors should introduce their term "aggregation index" more clearly. I understood that they count as an aggregate a singular pattern with a pixel number > X and a summarizes brightness >Y. I couldn't find the definition for a minimum size of aggregate. Not needed?

Figure 5:

c) The LB domain (aa170-220) is about 5kDa, a single FLAG tag is about 1kDa in size. Thus, WT-p62 with a single FLAG-tag should run at 63kDa and with an LB deletion at 58kDa. However, the WT has been marked with 70kDa, which would be too big. d) A very nice experiment, but why not using untagged UXT (Western blots of eluted fractions)? I myself find it difficult to rely on an experiment when the tag is much bigger than the fusion partner. Could also be done with purified UXT protein to

exclude unwanted binding of other proteins. f) Again, nice experiment, but why using the mCherry having good UXT antibodies?

Figure 6:

a) The authors should use the term "significant" only when they quantified and evaluate their analyses using statistics (page 11, second line; page 11, line 11). I disagree that the L/A mutant behave similarly, the L/P mutant is much stronger obviously (a, b). e) I miss the scan for WT-UXT.

Figure 7:

This is a nice experiment, which complements the data. Though ALS starts usually at the age of 50/60 in humans, the strong overexpression of mutant SOD1 may already affect tadpoles at an advanced stage when nerve-muscle connections are established. NF stage 28 of *Xenopus laevis* is the first stage of rudimentary swimming (escape swimming), nicely described by Louise Muntz in 1975. To my knowledge, it is not clear whether the motoneuron already fire, or swimming (muscle cell contraction) is a more autonomously activity at NF stage 28. It is interesting to see that the co-injection of synthetic RNA (I wouldn't call it mRNA) encoding UXT can rescue the effect of mutant SOD1, while mutant UXT cannot. But at the same time, this is a little bit surprising, since the LB domain of p62 is not conserved in *Xenopus*. Thus, I miss an experiment showing that human UXT can still bind *Xenopus* p62 (similar to supplementary figure 4).

REVIEWER COMMENTS

Reviewer #1 (Remarks to the Author):

In the manuscript by Yoon, M. J. et al. the authors show that the prefoldin-like protein UXT interacts with the autophagy cargo receptor p62 through p62's LB domain and thereby promotes SOD aggregate removal. UXT co-localizes with aggregated p62 or occasionally LC3 in cellular punctae and assists in the clearance of p62 bodies via autophagy. The authors hypothesize that the hexameric state of UXT helps cross-linking p62 aggregates inside cells as disruption of this oligomeric state reduces aggregate removal. Interestingly, aggregate removal by a p62 truncation lacking the PB1 domain responsible for homo-oligomerization can be rescued by UXT transfections. Finally, Yoon et al. show evidence that UXT overexpression in *Xenopus* tadpoles prevents SOD aggregate accumulation in neurons and thereby prevent loss of motor ability.

This manuscript demonstrates for the first time that the UXT co-chaperone interacts with p62 and is involved in aggregate removal. It covers a broad range of different cellular methods and model systems including cell lines as well as tadpoles. The manuscript has the potential to be published, once the following points are addressed:

We thank the reviewer for the constructive criticisms and useful comments. Based on the very helpful comments from the reviewer, we were able to strengthen our manuscript.

Major points:

1. The authors show that UXT transfection mediates the clearance of SOD aggregates (Figure 5d). However, it has also been shown that prefoldin co-assembles with chaperonins such as TriC (PMID: 30955883). The authors make the claim that UXT's activity of aggregate removal is due to autophagy. Nevertheless, chaperones with UXT's co-chaperone activity should also be considered as a contributing factor to dissolve aggregates. For instance, as a control it would be helpful to see the aggregate removal effect mediated by other chaperones such as Hsp70 and TriC.

Response: We appreciate the reviewer's helpful suggestion. As the reviewer's suggestion with the paper regarding prefoldin, the UXT's closely related protein, we tested UXT's co-chaperone activity in our experimental setting with various ways as below.

1) As the reviewer suggested, we investigated whether Hsp70, a chaperone that may not have an autophagic function, also promotes the removal of SOD1(A4V) aggregates. In case of TriC, it was not possible to compare its effect directly with UXT, because TriC consists of 8 different subunits that are difficult to be expressed in correct stoichiometry by co-transfection. In the experiment, we observed that overexpression of Hsp70 did not reduce SOD1(A4V) aggregates as efficiently as UXT (sFig. 3h; page 10, line 5).

2) To test the importance of UXT as a co-chaperone as suggested for prefoldin's function targeting aggregates into chaperonin (PMID: 309557883), we inhibited the function of chaperonin activity by using a potent Hsp70 family inhibitor, VER-155008. We found that UXT can reduce the aggregates even in the presence of the inhibitor (sFig. 3i; page 10, line 6), while inhibition of autophagy can block the effect of UXT (sFig. 3i and Fig. 4b).

3) Finally, we asked whether p62, the essential linker between UXT and autophagy, is required for the protective effect of UXT. Indeed, we found that knocking down the expression of the endogenous p62 completely blocks the ability of UXT to prevent SOD1(A4V)-induced motor neuron degeneration in *Xenopus* (Fig. 7i and 7j; page 14, line 20), strongly suggesting that autophagy plays important roles for the UXT's protective function.

Together, the results above support and strengthen our original hypothesis that UXT's ability to promote clearance of toxic aggregates is mainly mediated by its function in autophagy. Consistently, in our originally submitted results (Fig. 4b, 4c, and 4e), we observed that UXT-induced reduction in SOD1(A4V) aggregates was blocked by an autophagy inhibitor, bafilomycin.

However, as the reviewer pointed out, we do not rule out the possibility that UXT has a chaperon function. To make this point clear, we also added the sentence, "although we do not exclude the possibility that UXT can also act as a chaperone refolding misfolded proteins in a context-dependent manner" (page 10, line 8).

Additionally, does the UXT L/P mutant show that this is a specific effect due to p62 binding and autophagic clearance?

CK: It seems that the reviewer meant UXT L/A mutant but not UXT L/P mutant, because it was L/A mutant that has a defect in p62 binding activity (Fig. 6a) but not in other properties of p62 (Fig. 6d-f). Due to its specific defect only in p62 binding, the L/A mutant was used to suggest the importance of p62-UXT interaction in the UXT's protective effect against proteotoxicity in our original manuscript.

As the reviewer pointed out, however, this type of mutation study cannot completely exclude other possible unknown defects of the mutant. Therefore, to confirm the involvement of p62 in the UXT-dependent protective effect, we performed p62 knock-down in *Xenopus* as described above and demonstrated that UXT's protective function depends on the existence of p62 (Fig. 7i and 7j). Accordingly, the manuscript has been updated (page 14, line 20). We deeply appreciate this comment which led us to design a better way to prove our point.

2. In order to demonstrate that UXT is involved in autophagy, the authors essentially use an autophagy flux assay by inhibitor Bafilomycin. Currently, Figure 3a only shows only an increase in UXT levels. Please include p62 as well as LC3 Western blots. LC3IàLC3II conversion alongside p62/UXT increase will make the point clearer and point strongly to autophagy. Quantification of all band intensities will be helpful to verify bona fide autophagy turnover. This type of standard autophagy flux assay is generally recommended by a community consensus paper (Klionsky et al., 2016 Guidelines for the use and interpretation of assays for autophagy).

Response: We thank the reviewer for the suggestion. As suggested, we have performed the experiment including p62 and LC3 western blots again (Fig. 3a). We've also included the quantification data (Fig. 3b).

3. It is notable that very little p62 Δ LB in comparison to wt-p62 can be found in the so-called insoluble fraction, independent of UXT addition (Figure 5c). Please quantify protein amounts in the different lanes, while correcting the varying amounts in loading control. Upon LB domain removal p62 filament formation or p62 interaction with other proteins is already significantly affected. No explanation is offered in the text. Please clarify as this p62 Δ LB mutant is used throughout the manuscript.

Response: As requested, we've included quantification data of the detergent-insoluble p62 normalized against total p62 and updated the figure accordingly (Fig. 5d). In addition, we've included a possible explanation on the reduced localization of p62 Δ LB at the insoluble fraction (page 10, line 17); "LB domain removal may affect intramolecular and/or intermolecular interactions for p62 body formation. Because the LB domain was identified as the UXT binding site, we assumed that UXT-p62 interaction might be one of such interactions needed for efficient p62 body formation."

4. The discussion section of the manuscript should be heavily reworked:

- a) First, it is very short and should be extended. There is a whole body of work regarding aggrephagy that could be put into context.
- b) Second, there is also more known on the mechanisms of prefoldin as a co-chaperone.
- c) Third, the hypothesis of p62 weaving is not easy to follow and should be highlighted as speculation.

Response: We appreciate this helpful suggestion. We've extended discussion section by including various research on aggrephagy (page 17, line 3~12). We've also included the co-chaperone function

of prefoldin (page 4, line 14~16). Finally, we have tried to explain better the hypothesis of p62 weaving (page 18, line 2~8) and added a hypothetical model (supplementary Fig. 8b). We appreciate the reviewer's helpful inputs.

5. The following conclusion is unclear and not justified by the results:

page 7: “If this is the case, the interaction between the hydrophobic surface of UXT and misfolded proteins under pro-aggregation conditions may prevent degradation by UPS, and promote autophagy via the recruitment of p62 to the UXT-aggregate complex.” The accumulation of UXT upon MG132 treatment is consistent with UXT removal by the UPS. It is far-fetched to state that the hydrophobic interaction is critical here as it has not been demonstrated by the experiments.

Response: We thank the reviewer for pointing our overstatement. In the revised manuscript, we've removed the sentence and just stated the possible role of UXT under an aggregation-prone condition (page 8, line 8). The figure regarding the statement (Fig. 3c in the original manuscript) has been also moved to supplementary Figure 8a and just used as a discussion.

Minor points:

1. Related to major point 1, please include more background information on UXT in the introduction section.

Response: In the revised manuscript, we've included the information on UXT in terms of its possible chaperone function related to its closely related chaperone, prefoldin. (page 4, line 10 ~ 17).

2. It is very easy to miss that additional efforts have been made to identify the critical stretch of the p62 LB binding region. This data is hidden in the Supplementary Figure 4. It should find a more pronounced place close to Figure 1f. Please improve phrasing in the results section and name the relevant residue region to make the vague statement for concrete: “We subsequently used binding analysis with p62 deletion mutants to further map the UXT binding site to the N-terminal half of the LB domain (Fig. 1f and supplementary Fig. 4)”

Response: We have moved the data to Figure 1i-j, and clarified the region with relevant amino acid numbering (Page 6, line 6~9).

3. In Figure 1j, please include a p62 Western blot to verify the presence of p62 in the insoluble fraction.

Response: We've updated Fig 1j to include p62 blot as well as LC3 in insoluble fraction (now Fig 11 in the revised manuscript).

4. Figure 3c: Structural model of UXT including electrostatic surface plot. Please describe how this model was generated in more detail. Is it based on a homologous structure? If yes, based on which protein, what was the sequence identity and how was this model generated?

Response: In the revised manuscript, we've included details for generating the model in the legend for the figure (the legend for supplementary Fig. 8a). The figure has been moved to supplementary Figure 8a.

5. Figure 3d: The intensity profiles are not showing colocalization conclusively as they are the intensity overlays of one line in one cell, which makes it hard to judge whether this is statistically significant. A better way to depict this is to make a scatterplot, where UXT intensity is plotted against p62 intensity and a correlation coefficient is calculated and compared to p62 + LC3 and UXT + LC3.

Response: We highly appreciate the reviewer's suggestion. As suggested, we've employed the scatterplot (supplementary Fig. 3b), and presented Pearson's correlation coefficients with statistical analysis (Fig. 3e). The manuscript has been updated accordingly (page 8, line 17~19).

6. Figure 3f: The p62 and LC3 bands have very low intensity above background level. Please show a SEC chromatogram and an SDS-PAGE or native PAGE analysis of the different fractions. A ponceau stain of the blot would also be helpful. Presence of a large series of other proteins could indicate additional substrate targeting to the autophagosome.

Response: We've provided the SEC profile (Fig. 3f) and the coomassie-stained gel image (Fig. 3g). We've actually performed LC/MS analysis with the sample and identified many proteins found in intracellular membranes including ER and vesicles. We thought those proteins need to be analyzed further as an independent research.

7. Figure 7d and e. There is no unit of the y-axis. How is this 0 – 100 value obtained?

Response: The Y-axis indicates the percentage of tadpoles. We've added the information in the legends to the figures where necessary (e.g. the legend for Fig. 7d) in the revised manuscript. We appreciate the reviewer for careful reading of our manuscript.

Reviewer #2 (Remarks to the Author):

UXT is a prefoldin-like chaperone and Yoon et al used a yeast two-hybrid screen to identify UXT as binding to p62/SQSTM1. UXT oligomerizes and binds to p62 to increase the ability of p62 to form p62 bodies and degrade protein aggregates by autophagy. A *Xenopus* model of motor neuron degeneration caused by expression of the ALS SOD1(A4V) mutant is used to show that overexpression of UXT delays motor neuron degeneration. The L50A/L59A mutation of UXT which inhibits its binding to p62 is unable to prevent motor neuron degeneration in the *Xenopus* model.

I find this paper very interesting and bringing the novel feature of an oligomerizing chaperone binding directly to p62 to enhance the ability of p62 to degrade protein aggregates by autophagy. However, as detailed below, the data shown do not always justify the conclusions made and more control experiments are needed as well as better quantifications of some of the data. A major problem is the use of overexpression by transient transfections in some experimental settings when stable cell lines with more physiological relevant expression levels should be used instead.

Response: We appreciate the reviewer for acknowledging the novelty and importance of our study. We also thank the reviewer for his/her very critical comments and very helpful suggestions for improving our study. We've truthfully performed experiments suggested by the reviewer and hope our efforts can relieve the reviewer's concerns.

1. Fig. 1b: p62 is overexpressed (OE) by transient transfections in HeLa cells KO for p62. The OE of p62 appears quite high and it would have been better to use stable cell lines with close to endogenous expression level of WT and deletion mutant of p62.

Response: To address the reviewer's point, we have used lentivirus encoding GFP-fused p62 and p62 Δ ZZ-LB (and p62 Δ LB as well for the point #9) to transduce HeLa/p62KO cells, which were then sorted by FACS to match the levels of GFP fluorescence. Then, we confirmed by western blot that the levels of the GFP-tagged p62 proteins were similar and comparable to that of the endogenous p62 proteins (supplementary Fig. 2e-f). These cells were analyzed again for their ability to induce p62 clustering (supplementary Fig. 2g-h; page 5, line 19). The manuscript also has been revised to address these points (page 5, line 19; Materials & Methods section).

2. While appreciating the quantification method developed to measure what the authors call clustering index, the authors should also quantify the number and size distribution of the p62 bodies for WT and deltaZZ-LB in Fig 1 to reveal the difference in these parameters.

Response: As the reviewer suggested, we've quantified the number and size of the p62 bodies and updated the manuscript, and the results of these analyses (supplementary Fig. 2c-d; page 5, line 17) are consistent with our original analysis of the clustering index.

3. In Fig. 2 data from a cell death assay based on blebbing as a readout is shown. Blebbing structures are compatible also with other types of cell death, as apoptosis. The authors should clarify which kind of cell death is studied here. Analysis of blebbing is not enough and other assays are needed such as i.e. Hoechst staining of nuclear fragmentation, annexin V staining or flow cytometry using propidium iodide.

Response: According to the reviewer's suggestion, we've tried very hard to verify the nature of cell death related to the blebbing. The problems we encountered include

1) For the primary neuron culture, we must use the cell cycle inhibitor, 5-fluorodeoxyuridine, to limit the growth of non-neuronal cells. Because the inhibitor causes cell death of any proliferating cell, Hoechst, DAPI, and annexin V staining caused lots of background (from remaining debris of dead cells) which could not be discerned from the signal from neurons.

2) Due to the limited number we can get from mice, primary neurons are not good for flow cytometry analysis. More importantly, axons of those neurons seem to be damaged during detaching them from cell culture dishes for flow cytometry analysis, causing nonspecific staining of annexin V and PI.

As an alternative way to address this point, we asked whether the neurons with blebbing are positive for cleaved caspase-3 staining, a key molecular characteristic of the initial stage of apoptosis. For this, we performed time-lapse imaging cultured DRG neurons after treating MG132, fixed the culture when blebbings are visible, washed unbound cells, and performed immunocytochemistry using the antibody against cleaved caspase-3. We then registered the images of immunocytochemistry with the matching movies and found that all neurons with blebs are positive for cleaved caspase-3 while neurons without blebs are negative (supplementary movie 1c). The results of these new experiments show that the cell death we observed is apoptosis. These results are now included in the revised manuscript (page 7, line 11). We highly appreciate the reviewer for this useful comment.

4. Is p62 required for reduction of cell death upon overexpression of UXT in the model used in Fig. 2?

Response: We thank the review for pointing out this critical point. Figure 2 (using DRG neuronal culture model) and Figure 7 (using in vivo *Xenopus* model, related to the comment #12 of the reviewer) of our original manuscript show that overexpressing UXT protects neurons from aggregate-induced cell death, possibly by promoting autophagic clearance of the aggregates, a process that requires p62.

In order to address this point, we used the *Xenopus* model, because we think that it is more suited in this experiment for the following two points. First, it is possible to overexpress UXT and knock down p62 in the same neurons quantitatively, by targeted microinjection of UXT-encoding RNAs and p62 antisense morpholino oligonucleotides into blastomeres that will give rise to the central nervous system. In cultured DRG neurons, the co-infection rate of two lentiviral vectors (one for UXT overexpression and the other for p62 knock down) is low (~2.25% based on the fact that the single infection rate is usually ~15% in our system, making it very hard to get enough number of cells for the analysis), whereas in *Xenopus* over 98% of neurons will contain UXT RNA and p62 morpholino (Reviewer Figure 1b). Secondly, the readout is more robust in the *Xenopus* model. The DRG experiment would rely on the morphological analysis of time-lapse imaging of the co-infected cells, which makes it difficult to be used as an assay that involves multiple experimental groups. We think that the live imaging experiment is ideal for the initial cell biological analysis of neuronal death, as we used in our original Figure 2, but for the mechanistic studies that such as the experiment regarding the point addressed here, the *Xenopus* model would be more efficient in addressing the same point.

In this new *Xenopus* experiment, we found that p62 is indeed required for the UXT's ability to prevent SOD1(A4V)-induced motor function deterioration. Although knocking down p62 alone in *Xenopus* did not cause any obvious defects in their morphology (Fig. 7b-c) or motor behavior (Fig. 7h) up to stage 43 overexpressing UXT in p62-knocked down *Xenopus* did not rescue the SOD1(A4V) phenotype (Revised Fig. 7i and 7j). This result indicates that p62 is required for the UXT's function against proteotoxicity. The manuscript has been updated accordingly (page 14, line 20).

5. The blot in Fig 3a needs to be quantified. It does not seem very clear from the blot alone that UXT is stabilized by Baf, indicative of UXT being degraded by autophagy while the stabilization by proteasomal inhibition looks very clear.

Response: As suggested, we repeated and combined all experiments to generate bar graph (Fig. 3b). The effect of bafilomycin A1 is not that strong, but it was consistently observed.

6. Use of ATG5 or ATG7 KO cells should be done to determine if UXT accumulates when macroautophagy is inhibited.

Will UXT accumulate in the absence of p62?

Response: Although transient knock-down of p62 could increase the level of UXT which has been incorporated in the revised manuscript (Fig. 3c), the accumulation of UXT level was not that observed

in p62, ATG5- and ATG7 knock out HeLa cells as well as ATG5 knock-out MEF cells (Reviewer Figure 1) suggesting there may be a mechanism for maintaining the level of UXT. We have indicated this fact (page 7, line 25 ~ page 8, line 3). We hope this seemingly negative results do not mitigate our main conclusion derived from many other supporting experiments.

7. Fig 3d needs better quantification of colocalization with statistics.

Response: In the revised manuscript, we've calculated Pearson's correlation coefficients and presented the data as bar graphs with statistics (Fig. 3e and supplementary Fig. 3b).

8. The data in fig4e should be accompanied by western blots of the insoluble fractions at the relevant time points.

Response: As requested, we've performed the western blots to show the level of SOD1(A4V) as well as UXT in insoluble fractions (supplementary Fig. 3e-f). Consistently with the Fig. 4e, addition of UXT could reduce SOD1(A4V) in the insoluble fractions (lane 1 vs lane 2, lane 3 vs lane 4, lane 5 vs lane 6) in most time point. Blocking autophagy could attenuate the ability of UXT in reducing SOD1(A4V) in the insoluble fractions (lane 6 vs lane 12).

9. Figure 5a: Again p62 overexpression levels are too high. Dot size in HeLa cells are much bigger than endogenous p62. These experiments are better performed in stable cell lines.

Response: As in the response for the comment #1, we've generated stable cell lines (supplementary Fig. 2e-f) and analyzed dots again (supplementary Fig. 4b-c). The manuscript has been updated (page 10, line 16~17).

10. Are the big aggregates (p62 Δ PB1 + UXT-mCherry) in Fig. 5f degraded by autophagy?

Response: Because p62 Δ PB1 in the detergent-insoluble fraction was increased by Bafilomycin A1 treatment (supplementary Fig. 4f in the revised manuscript), we assume the big aggregates (presumably loosely packed) can be degraded by autophagy. We've updated the manuscript (page 11, line 22~25).

11. Fig 6b shows irregular shaped aggregates that do not look like the round p62 bodies. The aggregates depend on UXT interacting with p62 and this suggest they are formed by UXT and not by p62 Δ PB1.

Response: The absence of p62 self-oligomerization and/or the deletion of PB1 domain may account for the irregular shaped-aggregates (page 11, line 22~23). In addition, to reduce a possible artifact from whole deletion of PB1 domain, we performed the similar experiment using an oligomerization-defective p62(K7A/D69A) mutant and observed that the shape of p62 bodies made by p62(K7A/D69A) mutant were similar to that of wild type (supplementary Fig. 4g and 4h). The manuscript has been updated accordingly (page 11, line 25 ~ page 12, line 4).

12. How does knock down of p62 affect the swimming ability in the Xenopus SOD model?

Response: We thank the reviewer for this interesting question. Missense mutations in p62 are indeed linked to ALS/FTLD in human, and p62 loss-of-function exacerbates motor neuron degeneration in ALS models in mouse and zebrafish in line with our hypothesis that autophagic clearance of toxic aggregates, involving UXT-p62, is required for maintenance of motor neuronal health and function (page 17, line 13~16). In the original manuscript, we showed that increasing UXT expression prevents SOD1 mutant-induced motor neuronal degeneration, at least during the time we observed. As the reviewer suggested, we performed additional experiments to ask whether decreasing expression of endogenous p62 make the animal prone to loss of motor neuron and function. As described in our response to the comment # 4 above, we found that knocking down p62 on its own has a marginal effect on motor function in the time point used in the original manuscript (up to stage 43) (Fig. 7h and

supplementary Fig. 7b-c). However, knocking down p62 completely blocks overexpressed UXT's ability to prevent SOD1 mutant-induced motor function deterioration (Fig. 7i-j), indicating the UXT promotes the clearance of SOD1 mutant aggregation in a p62-dependent manner.

Minor:

1. In the Introduction there are references to several rather old review that should be replaced by more updated ones.

Response: As the reviewer suggested, we've updated several review papers (references 1,3,4,5,11, and 16). We thank the reviewer for the suggestion.

2. The introduction is very short and should contain some background information on prefoldin-like chaperones and UXT.

Response: We've added more background information of prefoldin and UXT (page 4, line 9~17). We thank the reviewer for the suggestion.

3. Is it known if UXT may be lost or mutated in ALS cases?

Response: To our knowledge, no UXT mutations have been linked to ALS. We think that the severe developmental defects in UXT hypomorphic *Xenopus* (Fig. 8c) might be one reason. Indeed, no UXT loss of function alleles are known in healthy human (https://gnomad.broadinstitute.org/gene/ENSG00000126756?dataset=gnomad_r2_1). Intriguingly, however, UXT was shown to interact with the *Als2* protein, whose mutations are linked to ALS. We added this point in Discussion of the revised manuscript (page 17, line 16~19).

4. On page 3 the authors write: "phagophore (the autophagic membrane structure)". This needs to be corrected to either (the forming autophagosome) or (isolation membrane).

Response: We appreciate the reviewer for the correction. The manuscript has been updated (page 3, line 17).

5. On p4, 5 lines from top: The role of the KIR domain is not unclear as it mediates binding to KEAP1 and this is modulated by phosphorylation in the KIR motif.

Response: We appreciate the reviewer for the correction. We added the role of the KIR domain in the revised manuscript (page 4, line 3~5).

6. Delete "were" in line 5 from top on p 9.

Response: It has been deleted (page 9, line 19).

7. Ref 37 is the same as ref 15 only that the journal name is missing.

Response: We have corrected the error. We highly appreciate the reviewer for careful reading of our manuscript.

8. The formal name of p62, SQSTM1 (sequestosome-1) should be given when first mentioning p62. Actually, this name was given by J. Shin who discovered p62.

Response: We have included SQSTM1 in the abstract.

Reviewer #3 (Remarks to the Author):

This study identifies UXT in a yeast two-hybrid screen as an interacting partner with the ZZ/LB domains of p62, an aggregate degradation mediator. The paper demonstrates a role for UXT in the degradation of ubiquitinated aggregates via autophagy through its interaction with p62. In addition, the study demonstrates a novel role for UXT which rescues a CNS defect phenotype in *X. tropicalis*.

Major weaknesses:

One weakness of the study is that all of the phenotypic data demonstrating the role of UXT in aggregate degradation relies on UXT over-expression rescue experiments, rather than knockout or knockdown of endogenous UXT. While it is interesting that over-expression of UXT can rescue various findings, including aggregation of proteins and cell death in the presence of MG-132, there should be some evidence of the function of endogenous UXT in these pathways, and not only the endogenous interaction between p62 and UXT shown in Fig. 1g/1h. The manuscript should not be accepted without results showing perturbation of endogenous proteins affect function.

Response: We thank the reviewer for the constructive and fair criticism on the weakness of our study in the original manuscript. In the revised manuscript, we addressed the reviewer's concerns mainly by performing new experiments that involve knocking down of endogenous UXT and p62 gene expression. The results of these experiments, which are summarized below, support and strengthen our original conclusion that UXT promotes clearance of toxic aggregates in a p62-dependent manner.

First, we performed the new experiments that show endogenous p62 and UXT interact (supplementary Fig. 2i) and SOD1(A4V) accumulates when endogenous UXT gene expression is knocked down (supplementary Fig. 3g).

Secondly, we showed that UXT mRNA is normally expressed in the lower motor neurons in *Xenopus* (both by RT-PCR and in situ hybridization) (Fig. 8a-b) and knocking down UXT expression by antisense morpholino oligonucleotides causes severe developmental defects that can be rescued by co-expressing morpholino-resistant form of UXT protein (supplementary Figure 7d).

Thirdly, we show that a partial knock down of UXT to the level that does not cause developmental defects makes the animal unable to cope with proteotoxicity. In embryos in which UXT gene expression is partially knock down so that they develop without any morphological defects and gain full motor function (Figure 8c-e), expressing the same amount of SOD1(A4V) that causes neurodegeneration in wildtype embryos was lethal (not shown), suggesting that adequate UXT level is important for resilience to proteotoxicity. Furthermore, overexpressing the wild type SOD1 protein, which marginally affect motor function maintenance in control animals (supplementary Fig. 7e), caused motor function degeneration in almost all embryos tested (Fig. 8e). These results indicate that endogenous UXT gene is required for the neuron to cope with proteotoxicity.

Finally, we show that endogenous p62 expression is required for UXT to function. Specifically, we show that knocking down endogenous p62 in *Xenopus* completely blocks the protective effect of UXT overexpression against SOD1(A4V)-induced toxicity (Fig. 7i and 7j; page 14, line 20~page 15, line 1), which strongly support our original hypothesis that UXT promotes clearance of toxic aggregates by interacting with p62.

We hope that all these results directly address the reviewer's concern regarding the roles of endogenous UXT and p62.

Additionally, the claim that UXT forms a hexamer with a molecular weight of about 200 is somewhat unfounded, since the peak at 200 kDa on the SEC may also be a complex or aggregate of UXT with other proteins.

Response:

As supporting evidences, we performed the suggested native gel experiment (as the answer for the question # 5) and detected very sharp and monodisperse band near 250 kDa (supplementary Fig. 4e; page 11, line 3~17). Moreover, we added new figure showing that even oligomerization defective p62 mutant, p62(K7A/D69A), can form the p62 cluster when UXT is co-expressed (supplementary Fig. 4e). These suggest that UXT may form an oligomer. In vitro measurement of oligomeric state of UXT, however, was not possible in our hand due to the difficulty of its purification (even though we tried a lot by intensive collaborations with structural biology labs). Therefore, we have changed the term hexamer into oligomer, and toned down our notion on the UXT oligomer throughout the manuscript. In addition, we have added the possibility that the peak can be generated by interacting with other proteins (page 11, line 16~17) as the reviewer suggested.

Because our study is mainly focusing on the protective role of UXT-p62 interaction against proteotoxicity, we believe that characterization of exact biochemical nature of UXT oligomer in a purified system is beyond our scope. We sincerely hope our current limitation on the direct biochemical evidence does not impair the significance and novelty of our key finding proven by many different experiments.

Minor weaknesses:

1. In Fig. 1f, can the authors speculate as to why the Δ ZZ mutant p62 binds to UXT better than the WT?

Response: Actually, we observed that not only UXT but also other protein that interacts with LB domain can bind better to Δ ZZ mutant. Currently, we speculate that existence of the globular ZZ domain (PMID: 30120248) next to the LB domain can sterically hinder the access of LB binders to the region. We have added this speculation in the revised manuscript (page 6, line 9~10).

2. In Fig. 2c, what is the role of endogenous UXT in the neuronal cells? Is it expressed? Over-expression of UXT results in improved cell survival in the presence of MG-132. Is this due to the presence of more UXT? How much over-expression is required to see this effect?

Response: In the DRG culture used in Figure 2, endogenous UXT expression is confirmed by staining (Fig. 2a). As the reviewer mentioned, the results in Fig. 2c indicates that overexpression of UXT prevents cell death induced by MG132. In the system, we think that enough UXT expression over existing protein aggregate would be the key for its protective effect. However, overexpression relies on lentiviral infection of UXT cDNA under UBC promoter with usually ~15% cells infected by single viral particle. This experimental setup limits controlling UXT expression quantitatively. Therefore, to address this question, we used *Xenopus* system in which overexpression of UXT prevents cell death by SOD1 mutant (Fig. 7 in the original manuscript), as it is possible to control the level of UXT expression by microinjection of morpholino or RNA.

We first confirmed that the lower motor neuron in the *Xenopus* spinal cord expresses UXT, both by RT-PCR and in situ hybridization (Fig. 8a-b, in the revised manuscript). For the quantification analysis of UXT expression, we used the quantitative knockdown of endogenous UXT, instead of introducing different amount of UXT together with SOD1 mutant, because it is more reasonable to control the endogenous UXT expression against external proteotoxic stress. In this experiment, we found that knocking down endogenous UXT using a translation-blocking antisense morpholino oligonucleotide that specifically blocks the expression of UXT (supplementary Fig. 7d) can induces development defects in a dose-dependent manner (Fig. 8c). When injected with 0.5 ng of UXT morpholino, approximately 40% of embryos showed no clear development defects (Fig. 8c) and those normally developed tadpoles showed little motor function deficits (Fig. 8d-e). In this condition, overexpressing SOD1 mutants cause their lethality (not shown). Moreover, overexpressing wild type SOD1, which

does not lead to the motor functional degeneration on its own in control tadpoles (supplementary Fig. 7e), resulted in a very severe phenotype with almost all tadpoles becoming completely paralyzed (Figure 7e), showing the importance of the existence of UXT against proteotoxicity. The manuscript has been updated accordingly (page 15, line 11~ page 16 line 3). We sincerely hope these explanation and experiments can relieve the reviewer's concerns.

3. For Fig. 3a the effect of the autophagy inhibitor on UXT levels is very modest.

Response: Although the effect is modest, it was consistent. To make this point, we provided the quantification data (Fig. 3b). In addition, we've provided the data showing increased UXT level by p62 knockdown (Fig. 3c), another way of inhibition of autophagy.

For Fig. 3d what do the dotted arrows show?

Response: In the original analysis, co-localization of p62, UXT and LC3 was analyzed in the region indicated with the dotted arrows for line profile analysis of their fluorescence intensities. As the reviewer indicated, we admit that the method was not clear enough. Therefore, in the revised manuscript, we calculated Pearson's correlation coefficients instead using whole cell area for the colocalization analysis (Fig. 3e and supplementary Fig. 3b).

4. In Fig. 4a, it still appears that UXT is forming aggregates in the SOD1 WT GFP cells. What can explain this? These clusters do not appear to be that different from the clusters in the SOD1(A4V) cells. Can the authors use their quantification system to show this difference?

Response: The image shown happened to be chosen from many other images showing dispersed staining of UXT in SOD1 wild type plus mCherry-UXT-transfected cells. Nonetheless, UXT staining in SOD1(A4V)-transfected cells are actually bigger and denser than those scarcely observed in SOD1-transfected cells. To make this clear as the reviewer suggested, we've provided quantification data (supplementary Fig. 3c-d). We appreciate the reviewer for critical reading of our manuscript.

5. In Fig. 5e, the authors need more evidence that the ~200 kDa peak is an oligomer of UXT and not also a complex with other proteins. If the >669 kDa may be a complex of UXT associated with other proteins, why is this not true for the other peak? One way to do this would be to show a silver stain of the 200 kDa peak and show that there is only one band at the size of UXT. Another alternative would be to show a non-denaturing gel Western blot with UXT antibody and show a band at this larger size. The data here is not enough evidence to claim that it is a hexamer.

Response: As we addressed above in the same point raised by the reviewer, we've tried very hard to address this issue. For the experiment the reviewer suggested, a silver staining of 200 kDa peak, we need to purify UXT protein, because the peak we observed was from direct loading of PBS-based cell lysates of GFP-UXT-transfected cells into analytic size exclusion column. This will result in many different proteins in the silver staining. As mentioned earlier, purification of UXT protein was not an easy task.

Therefore, we performed the alternative method that the reviewer suggested, the non-denaturing gel Western blot with UXT antibody. In the experiment, we got the essentially same data with the analytic gel filtration assay. In the native gel, UXT signal was observed as possibly large aggregates stuck on the well and at near 250 kDa marker while addition of SDS in the sample shows ~15kDa UXT (supplementary Fig. 4e). Even with this data, it was not possible to guess whether the UXT form a hexamer. As the reviewer would know better, the exact size cannot be estimated in the native gel because the migration rate will be largely dependent on its overall shape. Moreover, it is still possible that UXT may form a complex with other proteins in the native gel. Therefore, we've removed the notion on "hexamer". Instead, using other supporting evidences (e.g. supplementary Fig. 4g), we have just suggested that UXT may form an oligomer. The manuscript has been updated accordingly (page 11, line 3~line 17).

6. Can the authors please specify the expected monomeric size of UXT-GFP for Fig. 5e and Fig. 6e? It is a bit hard to tell which peak is meant to be the “monomer”. Since GFP is around 27 kDa, it is also hard to calculate how 200 kDa would be a hexamer.

Response: We carefully looked at the SEC profile again and found that there was a mistake indicating maker size in the SEC profile. First of all, the mistake has been fixed in the revised manuscript (Fig. 5f and 6f).

The expected size of GFP-UXT we used was around 40 kDa. However, not only because it is not easy to estimate the exact molecular weight of the peak near 200 kDa in SEC but also because the experiment cannot rule out the possibility of other protein's interaction with UXT, we have removed the term, hexamer, and toned down the claim on the hexamer, as mentioned earlier.

7. The authors state that the L50A/L59A but not the L50P/L59P mutant of UXT retains some aggregate targeting function in spite of the defect in p62 binding. However, if you look at the co-IP in Fig. 6a, the L50A/L59A mutant does retain some p62 binding. Is it possible that the difference in aggregate targeting is actually due to the total loss of L50P/L59P binding vs. the partial loss in L50A/L59A binding?

Response: As the reviewer noticed, L50A/L59A mutant is less severe than L50P/L59P mutant in terms of p62 binding (newly added supplementary Fig. 5b-c), which has been indicated in the revised manuscript (page 12, line 17~18). However, even though the L50A/L59A mutant has a defect in p62 binding, its targeting to aggregate (and its SEC profile) was same with wild type UXT (Fig. 6d, middle panel; newly added Fig. 6e). This means its ability to target to protein aggregates is independent of p62 binding. To clarify this thought, we have updated the manuscript (page 12, line 25 ~ page 13, line 2).

8. How do you know that UXT and SOD1(A4V) are both expressed when co-injected?

Response: We confirmed that two proteins, both by co-expression in cells and by co-injection into *Xenopus*, can be expressed efficiently (Reviewer Figure 2). In *Xenopus* microinjection experiments, the efficiency of co-expression of two mRNAs injected into the central nervous system-fated blastomere was over 98%. Out of 556 SOD1-A4V-EGFP expressing cells, 552 cells also expressed co-injected mCherry (along with UXT) (Revised Figure 7k and Reviewer Fig. 2b). We added this point in the revised manuscript (Page 9, line 4~5; Page 14, line 13~15).

9. The claim in the discussion that UXT could be part of a therapeutic strategy should be elaborated on – how can you enhance the function of a molecular chaperone therapeutically?

Response: We were thinking to increase the level of UXT by gene delivery or by inhibiting its UPS-mediated degradation (Fig. 3a) to increase the protein level, of course, once its UPS-dependent degradation mechanism is fully understood. We actually identified some E3-ligases by LC/MS analysis of the proteins co-purified with UXT, and we thought that the UXT-E3 ligase interaction might be a target of therapeutic strategy. We have updated the discussion accordingly (page 19, line 5~8). We thank the reviewer for giving us the opportunity to express our thought clearly.

Reviewer #4 (Remarks to the Author):

The work of Yoon et al. aims to provide novel insights into the impact UXT, a prefoldin-type chaperone on the function of p62 in autophagy. The authors claim that the aggregation of proteins is prevented by the physical interaction of UXT with substrate helping p62 to eliminate accumulated protein, which otherwise can confer pathological cytotoxicity. Thus, the manuscript deals with an important and interesting question. In general, the technical quality and the presentation of the data is high. However, the work is not always solid in the present form. Nevertheless, the manuscript is of potential interest for the readership of Nature Communication after revision.

My concerns:

The author should provide the full western blots of their protein analyses, which include a proper size indication. A table of the commercial antibodies including the order numbers, dilution, and crossbench references showing their specificity would be helpful. Further, a table listing all expression constructs and their composition with the expected size of the resulting protein. The RAW data of their statistical analyses should be provided in the supplement. To me, it is not clear how the authors scanned & normalized their protein blots. How much protein has been loaded per sample? For the PCRs, the authors should provide the accession numbers of the transcripts analyzed and whether the primers target alternative splice forms, or in the case of overexpression can discriminate between the endogenous and exogenous transcripts.

Response: First of all, we thank the reviewer for acknowledging the potential impact of the study and providing helpful suggestions to improve our manuscript. As suggested by the reviewer, we have provided tables listing antibodies, reagent, primers, and constructs (supplementary information 1.xlsx), raw data for statistical analyses and the full western blots of our protein analyses (source data.xlsx). All ECL reactions for western blot images were taken using LAS 4000 without any saturated point, which was included in “Reporting summary” submitted. Methods for normalization protein blots are included in the figure legends, and the amount of protein loading (usually 15~20 µg of proteins) is in the Materials and Methods section.

Figure 1:

A tricky issue of the paper is that the identified UXT chaperone offers a sticky surface, which UXT uses to bind to unfolded or misfolded proteins. Though the yeast two-hybrid screen has been done properly in principle, it cannot be excluded that an even minor proportion of the bait (ZZ&LB domain of p62) is not folded correctly and thus forms an artificial binding surface for chaperones as UXT. In many YTH screens or protein IP/Mass spectrometry analyses, chaperones of different kinds depending on the complexity of the library, tissue, cell line are always in the top 10 candidates. Thus, the question of whether the physical interaction of UXT and (ZZ-LB)-p62 is real, needs even more attention.

Response: When we performed the Y2H screening, we’ve used two different regions of p62 as baits. One was the N-terminal half of p62 excluding PB1 domain (as indicated in Fig. 1), and the other was C-terminal half of p62 excluding UBA domain. Because UXT was exclusively identified only in the N-terminal half region and no other known chaperones were found as preys, we believe the interaction is specific. In addition, their physical interaction was confirmed by many biological and cell biological experiments. Genetic interaction between p62 and UXT has been also confirmed in new *Xenopus* study (Fig. 7i-j). We also performed additional experiments suggested by the reviewer as below, which we hope can relieve the reviewer’s concern.

Using co-IP of tagged-versions (HA and GFP) they show that HA-p62 and HA-p62deltaZZ pulled GFP-UXT but not delta-LB. Thus, the LB domain should contain the binding site for UXT. Here, I miss the IP of UXT, showing that p62 is bound in a complementary manner.

Response: We thank the reviewer for the suggestion. We carried out a reciprocal immuno-precipitation assay and confirmed that the LB domain is the binding site of UXT (supplementary Fig. 2j).

Moreover, co-IPs of endogenous UXT / p62 is essential, the respective antibodies are listed and used throughout the paper. For better control, the authors should employ their p62-KO cells in PLA, not showing just an IgG control. However, the PLA cannot fully replace the endogenous IP, since it is not surprising to find a chaperone close to protein aggregates with p62. It may simply indicate that both, p62 and UXT, indeed act together to clear protein aggregates, but they must not physically interact necessarily.

Response: The reason why we couldn't pursue the endogenous IP before was that endogenous UXT was not soluble under non-denaturing IP buffers that normally preserve protein-protein interaction. To address the reviewer's concern, we fixed the cells and conducted IP experiment using RIPA buffer which can solubilize endogenous UXT protein. The results have been added in the revised manuscript (supplementary Fig. 2i).

J. Protein's standard is missing.

Response: We've included protein's standard in all western data in the revised manuscript.

Figure 2:

It is an interesting observation that overexpression of (mCherry)-UXT seems to reduce the proportion of dying DRG neurons when aggregates cannot become degraded by the proteasome. WT-UXT is 18kDa and mCherry is 27kDa in size. Thus, the tag is much bigger than the tagged protein any may interfere with UXT activity/function (the big ball covers -even partially- the full activity/function of the small ball). An easy control would be to compare mCherry-UXT with an untagged version or with a much smaller tag (Flag).

Response: The experiment was based on the real-time imaging of DRG neurons infected with UXT-mCherry. To identify infected cells and analyze their behaviors, fluorescence tagging was inevitable. In addition, based on the newly added data showing that the developmental defects induced by knocking down endogenous UXT is almost completely rescued by mCherry-tagged UXT (supplementary Fig. 7d), suggesting that the tagging itself seems to cause, even if there is, negligible differences in its function. We hope this can address the reviewer's concern.

I wonder whether UXT blocks the progression of the cell cycle in your experiments. Thus, a comparison of the cell cycle state upon UXT treated vs. WT could be interesting.

Response: The DRG neurons used in this experiment are post-mitotic cells. In fact, the culture medium contains Uridine/Fluorodeoxyuridine to intentionally prevent cell proliferation of non-neuronal cells. Therefore, we expect that expression of UXT did not affect cell cycles in this setting.

Figure 3:

The authors present data, which show that p62 and UXT are close to autophagosomes at the same time. Please, indicate the secondary antibodies used in the legend of figure 3 (and in general).

Response: We've updated figure legends to indicate the secondary antibodies used for staining (e.g. Figs. 2a, 3d, 5a, and so on).

The Western blot in f shows three fractions of the elution of a 10His-eYFP-UXT version, the

resulting fusion protein has an estimated size of 45kDa but the blot indicates 15kDa.

Response: We appreciate the reviewer's careful reading our manuscript. The construct we used contains a HRV3C protease cleavage site between UXT and eYFP, which was cleaved during purification step. To clarify this fact, we've updated the purification method in the figure legend (Fig. 3f-h). Briefly, after purification using Ni-NTA column, the eluates were treated with His-tagged HRV3C protease and then passed through Ni-NTA column again. The flow-through from the second Ni-NTA column was then further separated by size exclusion chromatography as shown in Fig. 3g.

LC3B-II normally runs a little bit faster (smaller) than 15kDa, the authors should check. The author should clarify this discrepancy showing the raw data.

Response: We highly appreciate the reviewer's comment. We've checked again and found our error in labeling molecular weight marker. The molecular weight marker we originally labelled as 15 kDa was actually 10 kDa. We have submitted the raw data (source data.xlsx).

Again, the authors should explain why they didn't try to detect the endogenous proteins only (e.g. native IP of UXT followed by mass spectrometry).

Response: As the response above, endogenous UXT protein is not soluble in the IP buffer that normally preserves the interaction. Therefore, we performed IP experiment using fixed cells, and have provided an evidence of endogenous p62 and UXT interaction (supplementary Fig. 2i). As mentioned above, their physical interaction was also confirmed by their genetic interaction in *Xenopus* (Fig. 7i-j).

Figure 4:

The authors show that inhibition of autophagy rescues UXT induce degradation of mutant SOD1 aggregates. In figure 4a SOD1-WT also forms clusters (weak green signals), which are of the same pattern as mCherry-UXT. Is this an artifact of overexpressed SOD1? Shouldn't we expect SOD1-WT protein in mitochondria?

Response: The image shown was randomly chosen from many other images, in most case, showing dispersed staining of UXT in SOD1 wild type plus mCherry-UXT-transfected cells. To better demonstrate what we observed, we've provided quantification data (supplementary Fig. 3c-d).

In the text (page 9) it must-read ... “ (using a live cell imaging system Fig. 2d)” not 2b. The authors should introduce their term “aggregation index” more clearly. I understood that they count as an aggregate a singular pattern with a pixel number > X and a summarizes brightness >Y. I couldn't find the definition for a minimum size of aggregate. Not needed?

Response: We were referring to our live cell imaging system illustrated in Fig. 2b. To make it clear, we modified the sentence (page 9, line 16).

For the image analysis, we first calculated the mean fluorescence intensity (MFI) and the standard deviation of fluorescence intensities (SD) of the pixels in within an image of manually selected single cell. Next, if the fluorescence intensity of a pixel in the image of single cells is greater than $MFI + 3 \times SD$, then the pixel was defined as aggregates. When pixels defined as aggregates are displayed as white on the black background, the black and white images shown in Fig. 4d or supplementary Fig. 2a (right panel) can be generated. For calculating aggregation index of the chosen cell, we summed up the fluorescence intensities of pixels defined as aggregates and divide the sum with the MFI. As suggested, we added this explanation to the legend for supplementary code 1.

In our fluorescence images analyzed by this approach, there were few pixels selected individually as a single pixel). Even if it happened, the effect of the single pixel on the aggregation index would be

minimum because the methods do not count the number of aggregates but calculate the sum of total fluorescence intensities of the aggregates.

Figure 5:

c) The LB domain (aa170-220) is about 5kDa, a single FLAG tag is about 1kDa in size. Thus, WT-p62 with a single FLAG-tag should run at 63kDa and with an LB deletion at 58kDa. However, the WT has been marked with 70kDa, which would be too big.

Response: The construct has 3xFLAG. Those blots in the figure were skewed a bit so that those bands appear too close to 70kDa. We've fixed the error in the revised manuscript (Fig. 5c) and submitted the original image (source data.xlsx).

d) A very nice experiment, but why not using untagged UXT (Western blots of eluted fractions)? I myself find it difficult to rely on an experiment when the tag is much bigger than the fusion partner. Could also be done with purified UXT protein to exclude unwanted binding of other proteins.

Response: First of all, we thank the reviewer for appreciation of the result. As the reviewer would already know well, the SEC profile is based on the very sensitive detection of GFP fluorescence from small volume (50µl) of soluble GFP-UXT in detergent-free cell lysates. Therefore, without its detection using fluorescence, it is not possible to guess where the corresponding peaks are, making it impossible to even collect sample fractions for western analysis. Moreover, because the number of sample fractions to be analyzed is far much limited in the gel-based analysis, it is also impossible to judge the monodispersity of possible peaks.

Therefore, to address the reviewer's point, we performed a native gel western blot instead, using the non-tagged UXT construct. In the experiment, we observed the UXT signal as large aggregates stuck on the well and at near 250 kDa marker (supplement Fig. 4e) which was very similar to the SEC profile. The manuscript has been updated, accordingly (page 11, line 10~17). We appreciate the reviewers' helpful suggestion.

For the purification of UXT, we've actually tried very hard to purify UXT in various system even with intensive collaboration with other structural biology labs, but eventually failed to purify the protein pure enough to investigate its biochemical properties *in vitro*. The difficulty made us to depend on the fluorescence-based SEC experiment.

f) Again, nice experiment, but why using the mCherry having good UXT antibodies?

Response: We first thank the reviewer for the evaluation of this experiment. To address the concern, we conducted similar experiments using minimally modified constructs, HA-tagged UXT and an oligomerization-defective p62(K7A/D69A) mutant (but not whole deletion of PB1 domain) and got the essentially same results (supplementary Fig. 4g-h). In this case, tagging was inevitable to make sure double transfection of both constructs.

Figure 6:

a) The authors should use the term "significant" only when they quantified and evaluate their analyses using statistics (page 11, second line; page 11, line 11). I disagree that the L/A mutant behave similarly, the L/P mutant is much stronger obviously (a, b).

Response: We appreciate the correction. For the term "significant" in the second line of page 11 in the

original manuscript, we quantified our immunoprecipitation results and provide the statistics (supplementary Fig. 5b-c). The term “significantly” in the line 11 of page 11 in the original manuscript was replaced by “Importantly” the term we actually intended (page 13, line 2).

We agree that L/P mutant is much stronger than L/A mutant. To make this point clear, we have added a sentence saying that “the L50P/L59P mutation is more severe than the L50A/L59A mutation” (page 12, line 17). In addition, to emphasize their difference in their binding to p62 and their abilities to target to insoluble fraction, we’ve also added quantification data (supplementary Fig. 5c, and Fig. 6e, respectively). The minor defect of the mutant compared to L/P mutant made us to use the L/A mutant for subsequent *Xenopus* studies as a p62 binding-defective mutant.

e) I miss the scan for WT-UXT.

Response: The SEC experiment shown in Fig. 5e in the original manuscript (now Fig. 5f in the revised manuscript) was actually done simultaneously with its mutants shown in Fig. 6e in the original manuscript (now Fig. 6f in the revised manuscript). Therefore, those experiment can be compared directly. We’ve added this point (page 13, line 4).

Figure 7:

This is a nice experiment, which complements the data. Though ALS starts usually at the age of 50/60 in humans, the strong overexpression of mutant SOD1 may already affect tadpoles at an advanced stage when nerve-muscle connections are established. NF stage 28 of *Xenopus* (*laevis*) is the first stage of rudimentary swimming (escape swimming), nicely described by Louise Muntz in 1975. To my knowledge, it is not clear whether the motoneuron already fire, or swimming (muscle cell contraction) is a more autonomously activity at NF stage 28. It is interesting to see that the co-injection of synthetic RNA (I wouldn’t call it mRNA) encoding UXT can rescue the effect of mutant SOD1, while mutant UXT cannot. But at the same time, this is a little bit surprising, since the LB domain of p62 is not conserved in *Xenopus*. Thus, I miss an experiment showing that human UXT can still bind *Xenopus* p62 (similar to supplementary figure 4).

Response: We thank the reviewer for expressing interest in this experiment and raising the critical point about p62-UXT interaction.

As the reviewer pointed out, stage 28 in *X. laevis* is the “early swimming stage”, when muscle contraction is beginning to be coordinated (updated in the page 13 line 25 ~ page 14 line 1). The tadpoles of this stage do, however, show a full touch-induced escape response, indicating that sensory-motor integration takes place, and we used this response as a readout of motor neuronal function. This assay led us to conclude that the neuromuscular junction normally forms and then deteriorates in SOD1 mutant-expressing tadpoles, indicative of neuronal degeneration rather than developmental defect. We thank the reviewer for acknowledging this point.

We also thank the reviewer for pointing out our incorrect use of the term “mRNA”. We injected RNA in vitro transcribed with cap analog to mimic endogenous mRNA. We clarified this in the revised manuscript (Materials and methods) and change the term into “in vitro-transcribed capped RNA” (throughout the manuscript).

The LB domain of p62, which is required for its interaction with UXT, does not appear highly conserved across animal species, as the reviewer pointed out. Therefore, we performed a new co-IP experiment, and the result indicates that *Xenopus* p62 indeed binds to human UXT (newly added supplementary Fig. 7a in the revised manuscript). A closer look at the amino acids of the LB2 domains showed that a stretch of 10 or so amino acids (containing RKKVKG, red highlighted box in supplementary Fig. 6b) are highly conserved, and the co-IP experiment with serial deletion mutants suggests that this region may be indeed required for p62 to bind UXT (Fig. 1i-j).

Finally, we performed a loss-of-function study to ask whether endogenously expressed p62 is required for UXT's protective function. Knocking down p62 on its own has a marginal effect on motor function during the time window used in the original manuscript (up to stage 43), but results in a gradual loss of motor function at later stages (e.g. stage 45) (supplementary Fig. 7b-c and Fig. 7h). Importantly, knocking down p62 completely blocks the overexpressed UXT's ability to prevent SOD1 mutant-induced motor function deterioration (Fig. 7i-j), indicating that UXT promotes the clearance of SOD1 mutant aggregation in a p62-dependent manner.

Together, the results of these new experiments strongly support our original hypothesis that UXT promotes clearance of toxic aggregates by interacting with p62. We have added these points in the revised manuscript (page 14, line 20 ~ page 15, line 1).

REVIEWERS' COMMENTS

Reviewer #1 (Remarks to the Author):

The authors have addressed all my raised concerns satisfactorily. The paper is ready for publication.

Reviewer #2 (Remarks to the Author):

The authors have revised their manuscript and addressed my comments in an excellent way.

Reviewer #3 (Remarks to the Author):

This manuscript is now acceptable for publication.

Reviewer #4 (Remarks to the Author):

The authors answered my questions and comments almost to my complete satisfaction. Though they have not been able to show the effects by using endogenous proteins, the additional data strengthen their arguments considerably. The manuscript can be published after minor corrections.

Typos:

page 7 apoptosis "maker" > marker?

page 14: the meaning of sentence unclear: ", we "treated" a translation blocking antisense morpholino..."

REVIEWERS' COMMENTS

Reviewer #1 (Remarks to the Author):

The authors have addressed all my raised concerns satisfactorily. The paper is ready for publication.

Response: We appreciate the reviewer's very helpful suggestions that have significantly improved our manuscript.

Reviewer #2 (Remarks to the Author):

The authors have revised their manuscript and addressed my comments in an excellent way.

Response: We thank the reviewer for the critical comments and helpful suggestions which made us to significantly improve our manuscript.

Reviewer #3 (Remarks to the Author):

This manuscript is now acceptable for publication.

Response: Thanks to the reviewer's constructive and fair criticism on the weakness of our original submission, we could provide more convincing results to support our conclusion. We highly appreciate it.

Reviewer #4 (Remarks to the Author):

The authors answered my questions and comments almost to my complete satisfaction. Though they have not been able to show the effects by using endogenous proteins, the additional data strengthen their arguments considerably. The manuscript can be published after minor corrections.

Typos:

page 7 apoptosis "maker" > marker?

page 14: the meaning of sentence unclear: ", we "treated" a translation blocking antisense morpholino..."

Response: We appreciate the reviewer for careful reading of our manuscript and helpful suggestions. We have corrected the errors (page 7, line 14 ("maker" to "marker") and page 14, line 16 ("treated" to "microinjected")).